# Reinforcement Learning for Reachability: Guaranteeing Asymptotic Optimality

**Amogh Palasamudram** [1]   **Jakub Svoboda** [2 3]   **Suguman Bansal** [1]   **Krishnendu Chatterjee** [2]

## Abstract

*Reinforcement learning* (RL) for *reachability specifications* is fundamental in sequential decision-making, yet theoretical guarantees remain less explored. A recent work achieves *asymptotic convergence* to optimal policies. However, this approach provides limited insight into convergence dynamics. In this work, we present an alternative approach that provides deeper theoretical insights into convergence. Our approach builds on *PAC learning* with assumptions. PAC learning guarantees near-optimal policies with high confidence in finite time but requires knowing internal MDP parameters like minimum transition probability. We argue that while these parameters are unknown in RL, they can be iteratively refined and estimated with increasing accuracy. By iteratively satisfying PAC conditions, we show that exact optimality can be achieved in the limit. Empirical evaluations on standard benchmarks validate our theoretical insights into convergence dynamics.

## 1. Introduction

Reinforcement learning (RL) (Sutton & Barto, 2018) is a framework for decision-making in unknown stochastic environments modeled as Markov Decision Processes (MDPs). Many practical problems (Collins et al., 2005; Andrychowicz et al., 2020; Kumar et al., 2020; Levine et al., 2016) can be expressed as reward-based objectives, which admit strong theoretical guarantees: (i) asymptotic convergence (Watkins & Dayan, 1992), where learned policies are optimal in the limit; and (ii) PAC guarantees (Valiant, 1984), where learned policies are near-optimal with high probability after finite samples. Practical implementations of these algorithms have achieved significant real-world impact.

Several recent works have focused on the RL under high-level specifications that can be expressed with *Linear Temporal Logic (LTL)* or related logical formalisms (Alur et al., 2026; Aksaray et al., 2016; Brafman et al., 2018; De Giacomo et al., 2019; Hasanbeig et al., 2018; Littman et al., 2017; Hasanbeig et al., 2019; Yuan et al., 2019; Hahn et al., 2019; Xu & Topcu, 2019; Jiang et al., 2020; Li et al., 2017; Icarte et al., 2018; Jothimurugan et al., 2021). LTL objectives describe complex temporal behaviors, including safety and liveness, that are hard to express as rewards. In addition, this focus is motivated by the expressiveness and modularity of temporal logic, but it also raises fundamental challenges for learning and analysis.

Compared to classical reward-based objectives, LTL specifications describe a more complicated class of behaviors, which makes them harder to learn. General LTL specifications are not learnable with PAC guarantees (Yang et al., 2022; Alur et al., 2022). The feasibility of PAC guarantees for LTL objectives assumes dependence on internal parameters of the MDP such as the minimal transition probability (Ashok et al., 2019), mixing time (Perez et al., 2023), expected distance (Svoboda et al., 2024), and the like. These parameters are typically unknown, limiting the applicability of PAC-style analysis.

Asymptotic guarantees for LTL objectives remain largely unexplored. To the best of our knowledge, (Le et al., 2024) is the only prior work that achieves asymptotic convergence for LTL specifications, doing so by converting LTL to limit-average reward objectives. While this establishes theoretical feasibility, the approach relies on external parameters disconnected from the original MDP structure, providing no insight into convergence dynamics or when optimality emerges. A direct approach working with MDP parameters for LTL specifications, i.e. one without reward conversion, is essential for understanding how the intrinsic structure of the MDP and specifications govern convergence dynamics. Such an approach provides deeper insights into practical algorithm design and enables implementations with observable convergence guarantees. This motivates fundamental questions: *Can asymptotic convergence be characterized directly in terms of intrinsic MDP parameters? Do asymptotic guarantees translate into observable convergence in practice?*

[1]Georgia Institute of Technology, USA [2]Institute of Science and Technology, Austria [3]Dartmouth College, USA. Correspondence to: Amogh Palasamudram <amogh.p.214@gmail.com>, Jakub Svoboda <jakub.svoboda@dartmouth.edu>.

*Proceedings of the $43^{rd}$ International Conference on Machine Learning*, Seoul, South Korea. PMLR 306, 2026. Copyright 2026 by the author(s).

In this work, we respond to these questions in the affirmative through a direct learning approach for *reachability specifications* in unknown MDPs (Section 3). Reachability forms a fundamental basis for RL with LTL specifications, LTL objective to reachability is well-established (Sickert et al., 2016; Baier & Katoen, 2008; Hahn et al., 2020b;a). Consequently, any learning algorithm for reachability induces a learning algorithm for LTL specifications.

Our method provides guarantees significantly stronger than asymptotic convergence 3). Specifically, with probability one, there exists a finite threshold $K_{\mathsf{opt}}$ beyond which only optimal policies are observed. Moreover, we explicitly characterize $K_{\mathsf{opt}}$ using intrinsic MDP quantities, through the minimum value separation $\varepsilon_{\mathsf{diff}}$ between optimal and suboptimal memoryless deterministic policies (Section 4).

To achieve these guarantees, we develop a staged direct learning algorithm that incrementally refines learning parameter estimates, constructs partial MDP models via simulation, and computes a memoryless deterministic policy using conservative transition estimates and bounded value iteration, with stage-wise PAC guarantees. As stages progress, the approximation tolerance eventually falls below the $\varepsilon_{\mathsf{diff}}$, ensuring exact optimality. Combining stage-wise PAC guarantees with a summable failure schedule yields almost-sure convergence and ensures that beyond stage $K_{\mathsf{opt}}$ only optimal policies are produced.

Empirical evaluation on standardized benchmarks demonstrates that these theoretical guarantees translate to practice, with optimal policies emerging rapidly (Section 5).

These contributions significantly strengthen prior results such as (Le et al., 2024), whose approach relies on reward-based formulations and provides only asymptotic convergence in value, without any characterization of the emergence of optimal policies. Moreover, asymptotic guarantees for reachability specifications are of fundamental significance beyond reachability itself. As established in Remark 2.1, reachability is the core computational primitive underlying all $\omega$-regular and LTL specifications, and any learning algorithm that achieves asymptotic guarantees for reachability immediately induces one for the full class of LTL and $\omega$-regular specifications. Thus, a principled algorithm for reachability convergence is not merely a theoretical milestone but a concrete foundation for practical RL systems that learn from rich temporal specifications.

**Related work.** Conversion of LTL to reachability specifications is well-studied for both known (Baier & Katoen, 2008) and unknown (Hahn et al., 2020a;b) MDPs, occurring via product construction with limit-deterministic Büchi automata (Sickert et al., 2016). (Hahn et al., 2020a) even presents a *simulatable* conversion of RL from LTL to RL reachability. These results allow us to focus on reachability

specifications as the fundamental problem of learning LTL objectives.

The intractability framework of Alur et al. (2022) establishes that there is *no* optimality- or near-optimality-preserving translation from reachability objectives into discounted-sum rewards in RL, ruling out any reduction between the two problem classes. Reachability optimizes the probability of reaching a target state: in terms of rewards, a trajectory receives reward 1 if it visits the target and 0 otherwise. Even with these trivial rewards, discounted-sum based RL (including standard RL algorithms such as Q-learning, $E^3$/RMAX and goal-conditioned RL) assigns a reward of $\gamma^k$ to a trajectory that visits the target state at position $k$ for the first time, and 0 otherwise, where $\gamma \in (0, 1)$ is the discount factor.

(Le et al., 2024) proposes the first asymptotic learning for LTL objectives by reducing LTL specifications to limit-average reward optimization on product MDPs, solved via sequences of discounted-sum problems with discount factors approaching one (Blackwell's Optimality). While this establishes the theoretical feasibility of learning reachability objectives without finite-sample guarantees, it has limitations. First, convergence analysis relies on external parameters (discount factors) disconnected from the original MDP structure, providing no insight into convergence dynamics or when optimality emerges. Second, the framework remains theoretical without implementation, leaving questions about applicability and empirical convergence unanswered.

(Majumdar et al., 2025) develops an algorithm for RL with LTL specifications providing *regret-free guarantees*. However, these are significantly weaker than asymptotic guarantees: even with zero failure probability, regret-free frameworks permit infinitely many non-optimal policies.

## 2. Preliminaries

**Markov Decision Process (MDP).** A *Markov Decision Process* (MDP) is a tuple $\mathcal{M} = (S, A, s_0, P)$, where $S$ is a finite set of states, $s_0$ is the initial state, $A$ is a finite set of actions, and $P : S \times A \times S \to [0, 1]$ is the transition probability function, where $\sum_{s' \in S} P(s, a, s') = 1$ for all $s \in S$ and $a \in A$. We denote $p_{\min}$ the smallest nonzero transition probability. We define $Av(s)$ to be the set of actions available in state $s$. We assume that $Av(s) \neq \emptyset$ for all states $s \in S$.

*Runs.* An *infinite run* $\zeta \in (S \times A)^\omega$ is a sequence $\zeta = s_0 a_0 s_1 a_1 \ldots$, where $s_i \in S$ and $a_i \in A$ for all $i \in \mathbb{N}$. Similarly, a *finite run* $\zeta \in (S \times A)^* \times S$ is a finite sequence $\zeta = s_0 a_0 s_1 a_1 \ldots a_{t-1} s_t$. For any run $\zeta$ of length at least $j$ and any $i \leq j$, we let $\zeta_{i:j}$ denote the subsequence $s_i a_i s_{i+1} a_{i+1} \ldots a_{j-1} s_j$. We use $\mathsf{Runs}(\mathcal{M}) = (S \times A)^\omega$ and $\mathsf{Runs}_f(\mathcal{M}) = (S \times A)^* \times S$ to denote the set of infinite and finite runs, respectively.

*Policies.* Let $\mathcal{D}(A) = \{\Delta : A \to [0,1] \mid \sum_{a \in A} \Delta(a) = 1\}$ denote the set of all distributions over actions. A policy $\pi :$ $\mathsf{Runs}_f(S, A) \to \mathcal{D}(A)$ maps a finite run $\zeta \in \mathsf{Runs}_f(S, A)$ to a distribution $\pi(\zeta)$ over actions. We denote by $\Pi(S, A)$ the set of all such policies. A policy $\pi$ is *memoryless* if it is of the form $\pi : S \to \mathcal{D}(A)$. A policy $\pi$ is deterministic if it is of the form $\pi : \zeta \in \mathsf{Runs}_f(S, A) \to A$, i.e. , there is an action $a \in A$ with $\pi(\zeta)(a) = 1$.

*Probability and expectation measures.* Given a finite run $\zeta = s_0 a_0 \ldots a_{t-1} s_t$, the *cylinder* of $\zeta$, denoted by $\mathsf{Cyl}(\zeta)$, is the set of all infinite runs starting with prefix $\zeta$. Given an MDP $\mathcal{M}$ and a policy $\pi \in \Pi(S, A)$, we define the probability of the cylinder set by $\mathcal{D}_\pi^{\mathcal{M}}(\mathsf{Cyl}(\zeta)) = \prod_{i=0}^{t-1} \pi(\zeta_{0:i})(a_i) P(s_i, a_i, s_{i+1})$. It is known that $\mathcal{D}_\pi^{\mathcal{M}}$ can be uniquely extended to a probability measure over the $\sigma$-algebra generated by all cylinder sets. We use $\mathcal{D}_\pi^{\mathcal{M}}$ to denote the distribution over infinite runs in $\mathcal{M}$ induced by the policy $\pi$ and the associated expectation measure is denoted by $\mathbb{E}_\pi^{\mathcal{M}}$.

**Formal Specifications and Reachability.** Formal languages can be used to specify properties about runs of an MDP. A language specification $\mathcal{L} \subseteq \mathsf{Runs}$ is a set of *desirable* runs in an MDP.

Given a formal specification $\mathcal{L}$, the value of a policy $\pi$ w.r.t. specification $\mathcal{L}$ is the probability of generating a sequence in $\mathcal{L}$—i.e.,

$$J_{\mathcal{L}}^{\mathcal{M}}(\pi) = \mathcal{D}_\pi^{\mathcal{M}}(\{\zeta \in \mathsf{Runs}(S, A) \mid \zeta \in \mathcal{L}\}).$$

Let $\mathcal{J}^*(\mathcal{M}, \mathcal{L}) = \sup_\pi J_{\mathcal{L}}^{\mathcal{M}}(\pi)$ denote the maximum value of $J_{\mathcal{L}}^{\mathcal{M}}$ for all policies $\pi \in \Pi(S, A)$. We let $\Pi_{\mathsf{opt}}(\mathcal{M}, \mathcal{L})$ denote the set of all optimal policies in $\mathcal{M}$ w.r.t. $\mathcal{L}$—i.e., $\Pi_{\mathsf{opt}}(\mathcal{M}, \mathcal{L}) = \{\pi \mid J_{\mathcal{L}}^{\mathcal{M}}(\pi) = \mathcal{J}^*(\mathcal{M}, \mathcal{L})\}$. In many cases, it is sufficient to compute an $\varepsilon$-optimal policy $\tilde{\pi}$ with $J_{\mathcal{L}}^{\mathcal{M}}(\tilde{\pi}) \geq \mathcal{J}^*(\mathcal{M}, \mathcal{L}) - \varepsilon$; we let $\Pi_{\mathsf{opt}}^{\varepsilon}(\mathcal{M}, \mathcal{L})$ denote the set of all $\varepsilon$-optimal policies in $\mathcal{M}$ w.r.t. $\mathcal{L}$.

*Reachability specifications* comprise an important class of formal specifications. Given a set of states $G \subseteq S$, let a *reachability specification* $\mathcal{L}(G) = \{\zeta = s_0 a_0 \cdots \in \mathsf{Runs}(\mathcal{M}) \mid \exists i \in \mathbb{N}, s_i \in G\}$ be the set of all runs in $\mathcal{M}$ that visit a state in $G$. We refer to $G$ by the target states/accepting states.

*Remark* 2.1 (Significance of reachability specifications). First, reachability is the most basic temporal specification, e.g., it corresponds to the class of open sets in the topological characterization of temporal specifications. Second, if we consider general $\omega$-regular specifications, which subsume LTL specifications, then parity specifications provide a canonical way to express them (Safra, 1988). For MDPs with parity specifications, the optimal value is computed as follows (Courcoubetis & Yannakakis, 1995): (a) the set of

states $X$ where the optimal value is 1 (almost-sure winning set) is computed; and (b) the optimal value is the optimal reachability probability to $X$. The almost-sure winning set of an MDP $\mathcal{M}$ only depends on the associated structure $G(\mathcal{M})$ and not the precise probabilities, and can be computed efficiently (in sub-quadratic time) with discrete graph theoretic algorithms (Chatterjee & Henzinger, 2011; 2014). Thus, for all $\omega$-regular specifications, the core task is to solve the optimal reachability problem. Third, LTLf goals (De Giacomo et al., 2013) are also expressed as reachability goals. Given that reachability is the basic specification, and it is a core problem as mentioned above, in the sequel, we only focus on reachability specifications.

**Reinforcement Learning.** In *reinforcement learning (RL)*, the standard assumption is that the set of states $S$, the set of actions $A$, and the initial state $s_0$ are known but the transition probability function $P$ is unknown. The learning algorithm has access to a *simulator* $\mathbb{S}$ which can be used to sample runs of the system $\zeta \sim \mathcal{D}_\pi^{\mathcal{M}}$ using any policy $\pi$.

We define a *reinforcement learning task* to be a pair $(\mathcal{M}, \mathcal{L})$ where $\mathcal{M}$ is an MDP and $\mathcal{L}$ is a specification for $\mathcal{M}$. The goal of an RL task $(\mathcal{M}, \mathcal{L})$ is to use a *learning algorithm* to produce an optimal or near-optimal policy w.r.t. $\mathcal{L}$ in a simulator of $\mathcal{M}$. A learning algorithm $\mathcal{A}$ is an iterative process that in each iteration (i) either resets the simulator or takes a step in $\mathcal{M}$, and (ii) outputs its current estimate of an optimal policy $\pi$. A learning algorithm $\mathcal{A}$ induces a random sequence of output policies $\{\pi_n\}_{n=1}^{\infty}$ where $\pi_n$ is the policy output in the $n^{\text{th}}$ iteration.

A learning algorithm is said to converge *asymptotically* (Watkins & Dayan, 1992) if it is probabilistically guaranteed to converge to the optimal policy.

**Definition 2.2** (Asymptotic Guarantees). A learning algorithm $\mathcal{A}$ is said to converge asymptotically for $\mathcal{L}$ if

$$J_{\mathcal{L}}^{\mathcal{M}}(\pi_n) \to \mathcal{J}^*(\mathcal{M}, \mathcal{L}) \text{ as } n \to \infty \text{ with probability } 1$$

In contrast, a learning algorithm is *Probably Approximately Correct* (PAC-MDP) (Kakade, 2003) if it is guaranteed to learn a near-optimal policy with high confidence within a finite number of iterations. For a formal definition, see Appendix A.

## 3. Asymptotic Convergence

This section presents our main algorithm for the asymptotic convergence of reachability specifications. The key feature is that the algorithm proceeds without any conversion to rewards, and is based only on internal MDP parameters.

## 3.1. Overview and Intuition

Reachability specifications present a fundamental challenge in RL: PAC learning is impossible when transition probabilities are completely unknown (Alur et al., 2022; Yang et al., 2022). However, a key insight unlocks a path towards asymptotic guarantees: PAC learning becomes possible when certain MDP parameters are known. For example, if we know the least non-zero transition probability $p_{\min}$, PAC learning algorithms exist (Ashok et al., 2019).

Our algorithm exploits this observation to transform an impossibility result into an asymptotic guarantee. While we cannot know $p_{\min}$ in advance, we can systematically *guess* increasingly accurate approximations. By combining these guesses with carefully chosen but increasingly smaller approximation factors and confidence errors, we achieve exact optimality in the limit.

The algorithm (Algorithm 1) operates in stages $k = 1, 2, 3, \ldots$, where three key parameters are set in each stage. These are: (a) our guess for the minimum transition probability $p_k = 1/2^k$, (b) the confidence error $\delta_k = 1/2^k$, and (c) the approximation factor $\varepsilon_k = 1/2^k$. In every stage $k$, our algorithm simulates the MDP $N_k$ times to learn a partial model of the true MDP. It then extracts the optimal policy $\pi_k$ in the partial model. Policy extraction involves a procedure called *Bounded Value Iteration (BVI) (Haddad & Monmege, 2017)*. The asymptotic guarantee follows from three key observations:

1. **From Theorem 3.1 (PAC Guarantee)**, once $k$ is large enough that $p_k \leq p_{\min}$ (at stage $K_{\mathsf{PAC}}$), from this point forward, we can ensure a PAC guarantee in each stage, i.e. we can ensure that there exists a number of simulations $N_k$ such that the optimal policy in the partial model in stage $k$ is $\varepsilon_k$-optimal with probability at least $1 - \delta_k$.
2. As $k$ continues to grow, $\varepsilon_k = 1/2^k$ shrinks exponentially. Eventually, $\varepsilon_k$ becomes smaller than the minimum separation between optimal and suboptimal memoryless deterministic policies $\varepsilon_{\mathsf{diff}}$. Then **from Theorem 3.2 (Exact Optimality)**, at stage $K_{\mathsf{opt}}$ where $\varepsilon_k$ falls below this separation gap, there are no policies "in between" optimal and suboptimal. Any $\varepsilon_k$-optimal policy must be exactly optimal. Combining this with Theorem 3.1, we get that $\pi_k \in \Pi_{\mathsf{opt}}$ with probability at least $1 - \delta_k$. Morever, Section 4 presents further analysis on $\varepsilon_{\mathsf{diff}}$ w.r.t. MDP parameters.
3. Finally, **from Theorem 3.3 (Asymptotic Convergence)**, since the total failure probability across all stages is finite ($\Sigma_{k=0}^{\infty} \delta_k$), by Borel–Cantelli Lemma, with probability 1, only finitely many stages can fail. Hence, eventually, all $\pi_k$ must be optimal with probability 1.

Please note that $K_{\mathsf{opt}}$ and $K_{\mathsf{PAC}}$ are simply analytical parameters. They are only needed for the proof analysis. They are never used as inputs for our algorithm.

## 3.2. Core Algorithmic Concepts

**Transition Probability Estimation.** As the algorithm simulates the MDP, it stores counters of state-action transition occurrences $\#(s, a, s')$ and $\#(s, a)$ to calculate the lower estimate of the transition probability. Given an error tolerance $\delta_P > 0$, Hoeffding bound (Hoeffding, 1963) gives the following:

$$\Pr\left[\left|P(s, a, s') - \frac{\#(s, a, s')}{\#(s, a)}\right| \geq c\right] \leq \delta_P\,,$$

where $c = \sqrt{\frac{\ln(\delta_P/2)}{-2 \cdot \#(s, a)}}$ is the *width* of the estimation error. The computation is given in Appendix B.

Then, the lower estimate of transition probabilities $\hat{P}$ is given by $\hat{P}(s, a, s') = \max\left\{0, \frac{\#(s, a, s')}{\#(s, a)} - c\right\}$.

These choices of $c$ and the lower-estimate $\hat{P}$ are crucial to the PAC guarantee in Theorem 3.1.

**Bounded Value Iteration (BVI).** When transition probabilities are known, the *optimal state values* are computed as $V(s) = \max_{a \in A} V(s, a)$ where $V(s, a) = \sum_{s' \in S} P(s, a, s') \cdot V(s')$ for $s \in S \setminus G$ and $V(s) = 1$ for $s \in G$. When transition probabilities are unknown, we use a well-established technique called *Bounded Value Iteration (BVI)* (Haddad & Monmege, 2017) to compute lower and upper estimates of the optimal state values, denoted by $L(s)$ and $U(s)$, respectively.

A crucial concept here is of *end-components*. An *end-component* (EC) $C$ is a set of state-action pairs $(s, a) \in S \times A$ in an MDP such that (a) for all states $t \sim P(s, a)$, there exists an action $b$ such that $(t, b) \in C$, and (b) the graph induced by the state-action pairs in $C$ forms a *strongly connected component*. An end-component $C$ is said to be *maximal* if there does not exist another end-component $C' \neq C$ s.t. $C \subseteq C'$. For example, the set $\{(s_1, a), (s_2, a)\}$ is a maximal EC in Fig 1. For all $(s, a) \in C$, we call $a$ a *staying action* in state $s$. For all $a \in Av(s)$ such that $(s, a) \notin C$, we call $a$ an *exit action* in state $s$.

*BVI in the absence of ECs.* BVI computes $L$ and $U$ iteratively. It initializes $L(s) = 1$ for $s \in G$ and $L(s) = 0$ for $s \in S \setminus G$, and $U(s) = 1$ for all $s \in S$. These values are iteratively updated using the following equations:

$$L(s) = \max_{a \in Av(s)} L(s, a) \text{ and } U(s) = \max_{a \in Av(s)} U(s, a)\,,$$

where we define the lower and upper bounds of the state-

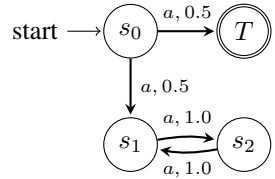

*Figure 1.* Simple MDP with EC

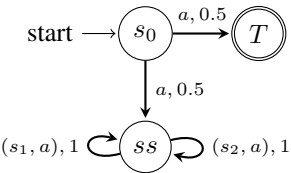

*Figure 2.* Collapsed MDP

action pair values as follows:

$$L(s,a) = \sum_{s':\#(s,a,s')>0} \hat{P}(s,a,s') \cdot L(s')$$

$$U(s,a) = \left( \sum_{s':\#(s,a,s')>0} \hat{P}(s,a,s')\, U(s') \right)$$
$$+ \left( 1 - \sum_{s':\#(s,a,s')>0} \hat{P}(s,a,s') \right)$$

These update equations, taken from (Ashok et al., 2019), guarantee that for all $s \in S$, $L(s) \leq V(s) \leq U(s)$ when we use lower estimates for transition probabilities.

*BVI in the presence of ECs.* In the presence of ECs, BVI may not converge. For example, in Fig 1, $C = \{(s_1, a), (s_2, a)\}$ forms an EC. Since neither of the states is a goal state, lower and upper bounds will initialize to 0 and 1, respectively. As per the iterative updates, these values will remain the same. This affects the convergence $L$ and $U$ in the initial state $s_0$, which will stabilize to 0.5 and 1, respectively.

To prevent this, each maximal EC (MEC) is *collapsed* to a single *super state*. In Figure 2, the MEC $\{(s_1, a), (s_2, a)\}$ is collapsed to a super state $ss$. The actions in the super state $ss$ are of the form $(s, a)$ where $s$ is a state in the MEC and $a \in Av(s)$. The transitions within the MEC are converted to self-loops on the super state. For example, in Fig 2, the transition $(s_1, a, s_2)$ is converted to $(ss, (s_1, a), ss)$. Transitions outside the MEC are retained with the new actions, i.e., $(s, a, s')$ becomes $(ss, (s, a), s')$. Transition probabilities are retained across all. Furthermore, for BVI to converge, for all $(s, a)$ in the MEC, we set $L(ss, (s, a)) = U(ss, (s, a)) = 0$ in the collapsed MDP if $G \cap MEC = \emptyset$. If $G \cap MEC \neq \emptyset$, then there is a target or goal state in the MEC. We would then set $L(ss, (s, a)) = U(ss, (s, a)) = 1$. A formal description of the collapsing procedure is in Appendix C.

BVI, as described previously, is run on the collapsed MDP till convergence. For instance, In Fig 2, the initialization of upper and lower bounds on $ss, (s_1, a)$ and $ss, (s_2, a)$ ensures that $L(ss) = U(ss) = 0$ upon executing the BVI iterations, eventually leading to $L(s_0) = U(s_0) = 0.5$.

Note, once we have collapsed the MDP, $PMC$, we *reset* all the upper and lower bounds for all states and state-action pairs. All upper bounds are reinitialized to 1. All lower bounds for state $s$ and state-action pair $(s, a)$ are set to either 0 if $s \notin G$ or 1 if $s \in G$. We do this to prevent previous stage errors from propagating to later stages.

**Policy extraction.** Once the lower and upper value estimates of the partial model are given by BVI, we can extract a memoryless deterministic policy. For a state $s$ that is not in an MEC, any action is chosen from its *best action set* where Best_Action(s) = $\arg\max_{a \in Av(s)} U(s, a)$.

We solve MEC $C$ as a separate reachability problem. For a state $s$ within an MEC $C$, if the state has a set of actions, $A' \subseteq Av(s)$ s.t. $A'$ is a subset of the $best\_exit\_actions(C)$, then any action from $A'$ is chosen. We treat such $s$ as a target in $C$; there is at least one such $s$, or the reachability is 0. Since some states in $C$ were changed to a target (with no outgoing actions), we solve the reachability on this simplified MDP recursively. Note that the value vector of a policy is the same if for other states that do not have the best exit action, we choose randomly $a \in Av(s)$ and $a$ is in the staying actions of $C$: Best_Exit_Action$(C)$ = $\arg\max_{a \notin stay\_actions(s)} U(ss, (s, a))$ where $ss$ is the super-state that $C$ collapses to.

A memoryless deterministic policy extracted in this manner is optimal w.r.t. the target set in the partial model.

### 3.3. Main algorithm

Our algorithm (Algorithm 1) runs in stages $k = 1, 2, 3, \cdots$ where each stage is parameterized by $p_k$, $\varepsilon_k$, and $\delta_k$. In every state $k$, the algorithm simulates the MDP to build its partial model and maintain counters for $\#(s, a)$ and $\#(s, a, s')$. We then compute lower and upper value bounds in the partial model using BVI, followed by optimal policy extraction in the partial model. For the sake of exposition, the pseudocodes are provided by ignoring parameters that are clear from context.

**Simulation.** Algorithm 2 describes the simulation procedure. A simulation terminates if the run either visits the target set $G$ or is stuck in an EC, as that indicates that the simulation is stuck in a cycle.

We use a simple heuristic (Algorithm 3) to detect whether an execution is stuck in an EC. The procedure stops a simulation when three conditions are met: the current state has

**Algorithm 1** RL from Reachability with Asymptotic Guarantees

```
 1: function ASYMPTOTIC(MDP M, target/goal set G)
 2:    k ← 0
 3:    PM ← ∅ {Current partial model}
 4:    repeat
 5:       k ← k + 1
 6:       δ_k ← 1/2^k, ε_k ← 1/2^k, p_k ← 1/2^k
 7:       for N_k times do
 8:          X ← SIMULATE(M, G) {Returns visited
             states}
 9:          PM ← PM ∪ X
10:       end for
11:       PMC ← COLLAPSEMECs(PM)
12:       for i = 1 to 2^k · |S| do
13:          BVI-UPDATE(PMC)
14:       end for
15:       π_k ← OPTIMALPOLICY(PM)
16:    until convergence
17: end function
```

been visited before (indicating a cycle), there exists a candidate set $C$ in the explored partial model that forms an EC, and the algorithm is $\delta_C$-sure that $C$ is truly an EC in the actual MDP. For simplicity, the explored partial model for EC detection is based on the counters $\#(s, a)$ and $\#(s, a, s')$ and not the estimated transition probabilities $\hat{P}$.

The $\delta_C$-sure EC detection provides a probabilistic guarantee without excessive sampling. If a state-action pair $(s, a)$ has an exit from the EC candidate $C$, that exit probability must be at least $p_k$, where we assume that $p_k$ is the minimum non-zero transition probability. After sampling $(s, a)$ for $n$ times, the probability of missing such an exit is $(1 - p_k)^n$, which should be smaller than $\delta_C$. This yields the required number of samples: $n \geq \frac{\ln(\delta_C)}{\ln(1-p_k)}$. The algorithm verifies that all staying state-action pairs in the EC candidate have been sampled at least $n$ times. A practical efficiency improvement is that the algorithm uses aggregated counters kept across all simulations rather than requiring $n$ samples within a single simulation.

Finally, with probability $1 - \mu$, the simulation procedure in stage $k + 1$ chooses the best actions according to the best actions determined in the previous stage $k$, for $k \geq 1$. Note, in the first stage, it chooses the best actions randomly from $Av(s)$ for all $s \in S$. With probability $\mu$, the simulation procedure in stage $k + 1$ chooses an action at random from $Av(s)$. $\mu$ is the exploration factor; it can be set as any arbitrary value, where $\mu \in \mathbb{R}(0, 1]$.

**Number of Simulations.** Next, we need to determine the number of simulations $N_k$ in each stage $k$.

We choose the number of simulations $N_k$ such that it ensures that the partial model (w.r.t. the estimated transition probabilities) is such that the optimal policy in the partial model is an $\varepsilon_k$ optimal policy in the original MDP with probability $1 - \delta_k$:

**Theorem 3.1.** *In Algorithm 1, there exists a $K_{\mathsf{PAC}} \in \mathbb{N}$ s.t. for all $k \geq K_{\mathsf{PAC}}$ there exists a number of simulations $N_k$ s.t. $\Pr[\pi_k \in \Pi_{\mathsf{opt}}^{\varepsilon_k}] \geq 1 - \delta_k$. I.e. policy $\pi_k$ is $\varepsilon_k$-optimal in MDP $\mathcal{M}$ with confidence $1 - \delta_k$.*

*Proof Sketch.* We present a proof sketch with the key ideas here. The complete rigorous proof has been presented in Appendix E.6.

Let the integer $K_{\mathsf{PAC}}$ be such that $p_{K_{\mathsf{PAC}}} = \frac{1}{2^{K_{\mathsf{PAC}}}} \leq p_{min}$ where $p_{min}$ is the minimum non-zero probability in the MDP $\mathcal{M}$. Note that even though the value $p_{min}$ is unknown (and never inputted into the algorithm), such a $K_{\mathsf{PAC}}$ will certainly exist. By running the algorithm continuously, we guarantee that stage $K_{PAC}$ is eventually reached.

Our objective is to show that we can bound the error in computing a near-optimal policy in all stages $k \geq K_{\mathsf{PAC}}$. The primary question is how many simulations do we require in each stage $k$ to bound the total error.

Let us call an action $a$ *relevant* in a state $s$ if the action is among the best actions in the partial MDP that can be chosen by the simulation in that given stage $k$. The corresponding transitions are called *relevant*. Intuitively, an action/transition is relevant if those are the actions the simulation would explore. A more formal definition of *relevant* action and transition can be found in Appendix E.2

Then (Ashok et al., 2019) proves that given $p$, a lower bound on the minimum non-zero transition probability, an approximation factor $\varepsilon$, and an arbitrary error $\delta$, there exists a number $s$ such that if each relevant transition is sampled $s$ times, we can guarantee learning a $\varepsilon$-optimal policy with error $\delta$. Extending this to our multi-stage scenario, when $k \geq K_{\mathsf{PAC}}$, we can guarantee the existence of such an $s_k$ for $p_k$, $\varepsilon_k$, and an arbitrary error $\delta_{TP}$ (note this may be different from $\delta_k$). Note, this works for EC-free MDPs. We include an extra $\delta_{EC}$ error to account for the detection and collapsing of MECs in the MDP.

Now, it remains to show that there exists a number of simulations $N_k$ such that each relevant transition is seen at least $s_k$ times.

We compute a (conservative) lower bound on the probability of seeing the least likely transition in a single simulation (including relevant transitions). We calculate this bound for all transitions, not just relevant ones, to account for the rare occasion the simulation does not sample from the *true* relevant actions in earlier stages; in this case, the algorithm

can adjust and correct itself by seeing all transitions enough times. We calculate this value, $p$, to be $p \geq \left(\frac{\mu \cdot p_k}{|A|}\right)^{|S|}$, as proved in Appendix E.4 Then, we will compute the number simulations $N_k$ required to visit the least likely relevant transition $s_k$ times. Note that visiting the least likely transition subsumes visiting all relevant transitions.

Let a random variable $X \sim \text{Binomial}(N_k, p)$. Then, we want to ensure that the cumulative distribution function of $X$ i.e. $F[X \leq s_k - 1] = \sum_{i=0}^{s_k-1} \binom{N_k}{i} \cdot p^i \cdot (1-p)^{N_k-i}$ is bounded, say by $\delta_{N_k}$, as this indicates the probability with which despite $N_k$ simulations, the least likely relevant transition has not been visited at least $s_k$ times.

To conclude, we compute the total confidence producing an $\varepsilon_k$-optimal policy after $N_k$ simulations are performed in stage $k \geq K_{\text{PAC}}$. There are three main sources of error: $\delta_{TP}$ from incorrectly producing a non-$\varepsilon_k$-optimal policy due to incorrectly estimating transition probabilities, $\delta_{EC}$ for incorrectly detecting and collapsing ECs, and $\delta_{N_k}$ from the failure to sample relevant transitions at least $s_k$ times. Hence, the error is $\delta_{TP} + \delta_{EC} + \delta_{N_k}$. We can choose $\delta_k$ such that $\delta_k = \delta_{TP} + \delta_{EC} + \delta_{N_k}$. Therefore, with $1 - \delta_k$ probability, stage $k \geq K_{PAC}$ will produce a $\varepsilon_k$-optimal policy. □

Next, we strengthen the previous argument to guarantee that the number of simulations $N_k$ is sufficient to learn optimal policies with high probability, as opposed to learning near-optimal policies with high probability. This utilizes our choice to decrease the approximation error $\varepsilon_k$ in each stage:

**Theorem 3.2.** *In Algorithm 1, there exists an integer $K_{\text{opt}} \in \mathbb{N}$ s.t. for all $k \geq K_{\text{opt}}$ there exists a number of simulations $N_k$ s.t. $\Pr[\pi_k \in \Pi_{\text{opt}}] \geq 1 - \delta_k$. I.e. policy $\pi_k$ is an optimal policy in MDP $\mathcal{M}$ with probability at least $1 - \delta_k$.*

*Proof.* In our algorithm, we consider only *memoryless deterministic policies* in $\mathcal{M}$. Note that (i) in the set of optimal policies, there exists a deterministic memoryless policy, and (ii) for a given $\mathcal{M}$, there are only finitely many memoryless deterministic policies.

Let $\varepsilon_{\text{diff}}$ be the difference between the value of the optimal memoryless deterministic policy $\pi^*$ and the first runner-up memoryless deterministic policy $\pi_2^*$ that does not achieve the optimal value. Note that since there are only finitely many policies, we have $\varepsilon_{\text{diff}} > 0$.

Let $K_{\text{opt}}$ be such that (a) $K_{\text{opt}} \geq K_{\text{PAC}}$ from Theorem 3.1 and (b) $\varepsilon_{\text{diff}} \geq \frac{1}{2^{K_{\text{opt}}}}$. Then, from Theorem 3.1, we get that for all $k \geq K_{\text{opt}}$, there exists an $N_k$ s.t. $\Pr[\pi_k \in \Pi_{\text{opt}}^{\varepsilon_k}] \geq 1 - \delta_k$. Note that $\varepsilon_k = \frac{1}{2^k} \leq \frac{1}{2^{K_{\text{opt}}}} \leq \varepsilon_{\text{diff}}$. Since $\pi_k$ is a memoryless and deterministic policy $\varepsilon_k$-optimal policy in

the original MDP $\mathcal{M}$, $\pi_k$ must be an optimal policy in $\mathcal{M}$. Hence, we get that for all $k \geq K_{\text{opt}}$ there exists an $N_k$ s.t. $\Pr[\pi_k \in \Pi_{\text{opt}}] \geq 1 - \delta_k$. Note, even though $K_{opt}$ is never inputted or known to the algorithm, it certainly exists. By running the algorithm continuously, we guarantee that stage $K_{opt}$ is eventually reached. □

A deeper analysis on $\varepsilon_{\text{diff}}$ has been presented in Section 4.

The final piece of our argument is to guarantee convergence to optimal policies with probability 1. So far, our procedure guarantees that beyond a stage, i.e., for all $k > K_{\text{opt}}$, the stage returns the optimal policy with an error of $\delta_k$. Generally, it is not sufficient to ensure convergence to optimal policies with probability 1 when each stage has a bounded error because the sum of these errors could diverge. However, in our case, the sum of these errors is guaranteed to converge to a finite value as we have carefully chosen $\delta_k = \frac{1}{2^k}$:

**Theorem 3.3.** *Algorithm 1 returns an optimal policy with asymptotic guarantees.*

*Proof.* Consider the infinite sequence of events $\{E_k\}_{k \geq 1}$ where $E_k$ is the event that the policy $\pi_k$ returned in the $k$-th stage is not an optimal policy. Let $K_{\text{opt}}$ be from Theorem 3.2.

Then, that sum of the probability of all events $E_k$ is finite $\Sigma_{k=1}^{\infty} = \Sigma_{k=1}^{K_{\text{opt}}} \Pr[E_k] + \Sigma_{k=K_{\text{opt}}}^{\infty} \Pr[E_k] = \Sigma_{k=1}^{K_{\text{opt}}} \Pr[E_k] + \Sigma_{k=K_{\text{opt}}}^{\infty} \delta_k \leq K_{\text{opt}} + \Sigma_{k=K_{\text{opt}}}^{\infty} \frac{1}{2^k} \leq K_{\text{opt}} + 1$. By the Borel–Cantelli Lemma, the probability that infinitely many events $E_k$ occur is 0. In other words, only finitely many events $E_k$ can occur with probability 1. Therefore, with probability 1, there exists a $K$ s.t. for all $k \geq K$, $\pi_k$ is optimal. In other words, our algorithm asymptotically converges to the optimal policy. □

Note that our guarantee is stronger than the general definition of asymptotic convergence. The general definition only requires the policies in each stage to keep getting arbitrarily close to the optimal policy with probability 1. This is, in fact, what (Le et al., 2024) proves. In contrast, our approach ensures that beyond a point, we produce optimal policies only with probability 1.

Furthermore, the stage beyond which this becomes possible, i.e., $K_{\text{opt}}$ depends only on $\varepsilon_{\text{diff}}$. In the following section, we argue that $\varepsilon_{\text{diff}}$ depends only on the internal parameters of the true MDP. As a consequence, if these parameters were known, we could even predict the exact stage beyond which the optimal-policy-only behavior is observed. We expand on this further in the next section.

# 4. Bound on $\varepsilon_{\mathsf{diff}}$

Theorem 3.2 shows that there exists a constant $\varepsilon_{\mathsf{diff}}$ such that the difference between an optimal policy and any non-optimal policy is at least $\varepsilon_{\mathsf{diff}}$. In this section, we give precise bounds on $\varepsilon_{\mathsf{diff}}$. Only for this section, we assume that all probability values are rational.

Our approach relies on techniques from linear algebra and adapts the approach of (Chatterjee et al., 2023) for MDPs with reachability. Technical details of the proofs and definitions are in Appendix F. We suppose that all the probabilities in the transition function are rational numbers. Moreover, we parametrize $\varepsilon_{\mathsf{diff}}$ by transition complexity, which can be viewed as of order of $1/p_{\min}$ (In theory, it can be higher than $\frac{1}{p_{\min}}$ if $p_{\min}$ is adversarially large).

We express the value vector of any policy as a solution to a matrix equation

$$v = v\mathbf{A} + \mathbf{r}, \tag{1}$$

where the $\mathbf{A}$ is square matrix with entries indexed by $S$ and $S \times A$. For policy $\pi$, states $s \in S \setminus G$, $s' \in S$ and action $a \in A$, we have $\mathbf{A}_{s,(s,a)} = 1$ if $\pi(s) = a$. We have $\mathbf{A}_{(s,a),s'} = P(s, a, s')$. Other entries of $\mathbf{A}$ are 0. For vector $\mathbf{r}$, on index $s \in S$, we have $\mathbf{r}_s = 1$ (or in general $w_s$, the reward for reaching $s \in G$) and 0 for all other indices.

**Transition complexity.** For one action-state pair, let the transition complexity for state $s$ and action $a$, $D_{s,a}$, be the smallest common multiple of all denominators of the transition function in the state $s$ after selecting action $a$. For the whole MDP, let the transition complexity $D$ be the maximum of the transition complexities of all state-action pairs, that is, $D = \max_{(s,a) \in S \times A} (D_{s,a})$.

With the formulation of the problem as a matrix equation and the knowledge of the transition complexity, we can prove the main theorem. The theorem provides a lower bound on the difference between optimal and non-optimal policies.

**Theorem 4.1.** *For all memoryless deterministic policies $\pi, \pi' \in \Pi$, we have $||v_\pi - v_{\pi'}||_1 \geq (2D)^{-2|A| \cdot |S|} 2^{-2|S|}$ or $||v_\pi - v_{\pi'}||_1 = 0$.*

*Proof Sketch.* We present a proof sketch. For details, refer to the Appendix F.

First, we construct the matrix $F$, a diagonal matrix, such that, after multiplying $A$ (from Equation (1)) by $F$, the entries of $FA$ are integers. Note that the entries of $F$ are bounded by $D$, the transition complexity.

Second, we express the solution of $vF(Id - \mathbf{A}) = F\mathbf{r}$ by using Cramer's rule. The rule states that the value of every variable is the ratio between the determinant of $F(Id - \mathbf{A})$ with some rows and columns omitted and the determinant of

$F(Id - \mathbf{A})$. However, the determinant of an integer matrix (recall that $F(Id - \mathbf{A})$ is an integer matrix) is always an integer.

Third, we provide the bounds on the determinant of $F(Id - \mathbf{A})$ that is at most $(2D)^{|A| \cdot |S|} \cdot 2^{|S|}$. Comparing two policies where the value vectors are expressed as fractions with denominators bounded by $(2D)^{|A| \cdot |S|} \cdot 2^{|S|}$ means that the values in one state are either the same for both states or different. In this case, the difference is at least $(2D)^{-2|A| \cdot |S|} 2^{-2|S|}$. $\square$

# 5. Implementation and Empirical Validation

Our theoretical contributions establish that Algorithm 1 achieves guarantees significantly stronger than standard asymptotic convergence: with probability one, beyond stage $K_{\mathsf{opt}}$ only optimal policies are produced. This section demonstrates that these theoretical guarantees translate to observable convergence in practice.

We developed a Python implementation [1] with practical adjustments to improve performance on real-world benchmarks while preserving the core algorithmic structure (Appendix G). We evaluate on 9 standardized MDP benchmarks from the Quantitative Verification Benchmark Set (Hartmanns et al., 2019). Each benchmark was evaluated over 10 independent trials (36-hour time limit, single CPU core, 1 GB memory, 2.4 GHz). The complete set of results and plots for all benchmarks are in Appendix H.

## 5.1. Empirical Results

Our experiments validate two key theoretical predictions.

**Emergence of optimal-policy-only behavior.** Theorem 3.2 establishes that beyond stage $K_{\mathsf{opt}}$, only optimal policies are produced with probability $1 - \delta_k$. Figure 3c demonstrates this for the Dining Philosophers benchmark: policy accuracy (empirical reachability probability) converges at stage $k = 2$ and remains stable thereafter. Across all benchmarks, policy accuracy converges at **median $k = 2$ and average $k = 2.3$**, confirming rapid emergence of the optimal-policy-only regime.

**Bounds convergence versus policy optimality.** A striking empirical observation is that optimal policies emerge *significantly earlier* than tight value bounds. While policy accuracy stabilizes at $k = 2$, bounds $L(s_0)$ and $U(s_0)$ only begin converging around $k = 16$ (Figures 3a-3b). This shows the algorithm finds optimal policies even when $U(s_0) - L(s_0)$ remains non-trivial, suggesting effective value separation exceeds worst-case $\varepsilon_{\mathsf{diff}}$ in practice.

---

[1] https://github.com/amoghp214/asymptotic-ltl-reachability

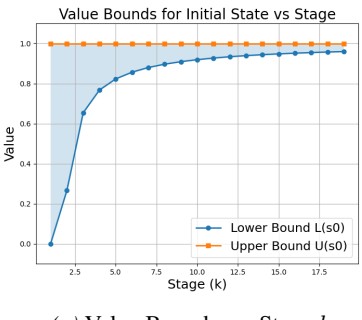 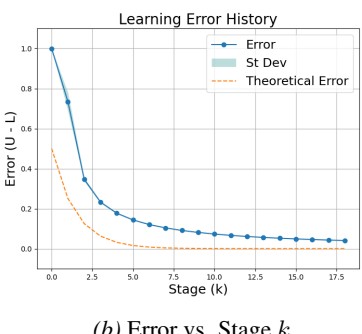 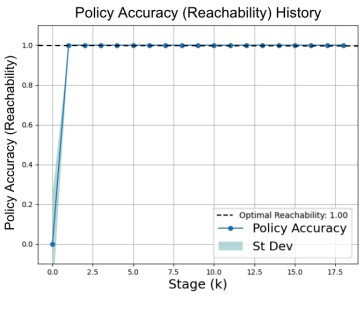

*(a)* Value Bounds vs. Stage $k$   *(b)* Error vs. Stage $k$   *(c)* Policy Accuracy vs. Stage $k$

*Figure 3.* Performance on Dining Philosophers benchmark. The values plotted are the median values over 10 trials. Optimal Policy Reachability Value: 1.00.

Figure 3a shows value bound convergence for Dining Philosophers (expected value $[1, 1]$): both $U(s_0)$ and $L(s_0)$ approach 1.0. Figure 3b compares theoretical sample size $N_k$ versus practical implementation, showing similar convergence profiles that validate our modifications preserve essential dynamics. Low standard deviations across trials (Figures 3b-3c) confirm robust convergence despite stochastic sampling variation.

Note, in Figure 3c, the policy accuracy is equivalent to the reachability probability of the learned policy. A low policy accuracy does not mean the algorithm performed poorly on the benchmark. The algorithm performs well if the policy accuracy converges to the true reachability probability of the optimal policy. These values can be seen in Table 1 in Appendix H. For example, the policy accuracy for the Zeroconf benchmark is $0$. However, the optimal policy's reachability probability is also $0$, meaning the algorithm works as expected.

## 6. Concluding Remarks and Future Work

We present the first direct learning approach for reachability that achieves guarantees significantly stronger than asymptotic convergence: with probability one, beyond a finite stage, only optimal policies are produced, where the stage is characterized through intrinsic MDP parameters. While our algorithm uses $p_{\min}$, the staged refinement framework can be generalized to other PAC guarantees under alternative assumptions such as mixing time (Perez et al., 2023) or expected conditional distance (Svoboda et al., 2024), opening opportunities for further theoretical analysis under different structural parameters.

Our current approach is model-based. Extending this framework to model-free methods—where policies are learned directly without explicit model construction—represents the critical path toward practical deployment and scalable RL for temporal specifications.

## Acknowledgment

Research reported in this publication was partly supported by an Amazon Research Award, Fall 2024. J.S. and K.C. were supported by the European Research Council (ERC) CoG 863818 (ForM-SMArt) and Austrian Science Fund (FWF) 10.55776/COE12. J.S. was supported by Dartmouth College.

## Impact Statement

This paper presents work whose goal is to advance trustworthy Machine Learning, in particular the, theory of RL under qualitative specifications. The contributions of this work are primarily theoretical and do not warrant any immediate societal consequence, to the best of our knowledge. There are many potential societal consequences of our work, none of which we feel must be specifically highlighted here.

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

# A. PAC Definition

**Definition A.1** (PAC-MDP). A learning algorithm $\mathcal{A}$ is said to be PAC-MDP for $\mathcal{L}$ if, there is a function $h$ such that for all $p > 0, \varepsilon > 0$, and all RL tasks $(\mathcal{M}, \mathcal{L})$ with $\mathcal{M} = (S, A, s_0, P)$, taking $N = h(|S|, |A|, |\mathcal{L}|, \frac{1}{p}, \frac{1}{\varepsilon})$, with probability at least $1 - p$, we have

$$\left| \left\{ n \mid \pi_n \notin \Pi^\varepsilon_{\mathsf{opt}}(\mathcal{M}, \mathcal{L}) \right\} \right| \leq N.$$

We say a PAC-MDP algorithm is *efficient* if the *sample complexity* function $h$ is polynomial in $|S|, |A|, \frac{1}{p}$ and $\frac{1}{\varepsilon}$.

## A.1. Splitting $\delta_k$

$\delta_k$ is broken up such that $\delta_k = \delta_{TP} + \delta_{EC} + \delta_{N_k}$.

**Probabilistic Error of Estimating Transition Probabilities** ($\delta_{TP}$)   $\delta_{TP}$ is the confidence error from estimating all the transition probability errors using the Hoeffding bounds. From this, we get the error for each transition's estimated probabilities:

$$\delta_P = \frac{\delta_{TP} \cdot p_k}{|SA|}$$

where $|SA|$ is the number of seen state-action pairs. Formally, for each transition $(s, a, s')$, $\delta_P$ is an upper bound for the probability that the value of $\hat{P}(s, a, s') > P(s, a, s')$, meaning $\hat{P}(s, a, s')$ is not the lower bound of the true transition probability, $P(s, a, s')$. In other words, with $1 - \delta_{TP}$ confidence, $\hat{P}$ is a lower bound for $P$.

**Probabilistic Error of Detecting End Components** ($\delta_{EC}$)   To account for probabilistic error in EC detection and MDP collapsing, $\delta_{EC}$ is the confidence error for determining whether the topology of the collapsed partial MDP is equivalent to the topology of the collapsed true MDP we are sampling from. Because EC decomposition and the collapsing process itself are deterministic, the only probabilistic error rises from the topology of the partial model being incorrect. An incorrect or incomplete learned topology could result in the EC decomposition algorithm (1) missing an SCC or (2) falsely collapsing an EC candidate because an unseen transition would make a current staying action and exit action. However, if the topology of the partial MDP is correct, EC decomposition will not fail. Thus, we use $\delta_{EC}$ error to account for topological errors. For the partial MDP topology to be incorrect, either there are unseen but reachable states or unseen transitions. Because unseen reachable states cannot be seen without unseen transitions, the problem boils down to checking for unseen transitions. The maximum number of possible transitions is $\frac{|SA|}{p_k}$, so the number of unseen transitions will be less than this value. Therefore, we can split $\delta_{EC}$ among $\frac{|SA|}{p_k}$ potential unseen transitions. Doing this, we see that

$$\delta_C = \frac{\delta_{EC} \cdot p_k}{|SA|}.$$

Now, for each potentially unseen transition, we must be $\delta_C$-sure that it does not exist. The probability a transition is unseen but exists is at most $(1 - p_k)^y$. Where $y$ is the number of samples taken for $\#(s, a)$. Hence,

$$\delta_C \geq (1 - p_k)^y.$$

To be rigorous, each state-action pair must take at least $y \geq \frac{\ln(\delta_C)}{\ln(1 - p_k)}$ samples. If an unseen transition, $(s, a, s')$ occurs, and $s' \notin S$, state $s'$ and its possible actions must be sampled $y$ times as well. This requirement is satisfied by the choice of $N_k$ as proven in Theorem 3.1. Hence, the probability error that the partial MDP is not collapsed properly is bounded by $\delta_{EC}$.

**Probabilistic Error of Seeing each transition $s_k$ times with $N_k$ simulations** ($\delta_{N_k}$)   $\delta_{N_k}$ is defined such that we are $1 - \delta_{N_k}$ confident that $N_k$ has been chosen such that the simulation phase runs $N_k$ times and every transition has been seen at least $s_k$ times. $s_k$ is defined and formalized in the proof of Theorem 3.1. $\delta_{N_k}$ is also split up into $\frac{|SA|}{p_k}$ parts. This is because there will be at most $\frac{|SA|}{p_k}$ transitions. Hence, for each relevant transition we say that there is an $\delta_n$ chance that it has not been seen at least $s_k$ times after $N_k$ simulations. $\delta_n$ is defined as

$$\delta_n = \frac{\delta_{N_k} \cdot p_k}{|SA|}.$$

## B. Hoeffding Bound

The two-sided Hoeffding bound (Hoeffding, 1963) claims that for a random variable $X$ where each $X_i$ is bounded by $[a_i, b_i]$

$$Pr[|\hat{X} - \mu| \geq \epsilon] \leq 2 \exp\left(\frac{-2n^2\epsilon^2}{\sum_{i=1}^{n}(b_i - a_i)^2}\right)$$

where $\bar{X}$ is the empirical mean, and $\mu$ is the true mean. In the case of transition probabilities, the empirical and true mean translate to the empirical and true probability of a transition's probability of occurring. All transition probabilities are bounded by $[0, 1]$, so $a_i = 0$ and $b_i = 1$ for all transitions. Replacing $\epsilon$ with $c$ (the confidence width), $\bar{X}$ with $\frac{\#(s,a,s')}{\#(s,a)}$, $\mu$ with $P(s, a, s')$, and $n$ with $\#(s, a)$. The inequality becomes

$$Pr\left[\left|\frac{\#(s,a,s')}{\#(s,a)} - P(s, a, s')\right| \geq c\right] \leq 2 \cdot \exp\left(\frac{-2\#(s,a)^2c^2}{\sum_{i=1}^{\#(s,a)}(1-0)^2}\right)$$

$$= 2 \cdot \exp\left(\frac{-2\#(s,a)^2c^2}{\#(s,a)}\right)$$

$$= 2 \cdot e^{-2\#(s,a)c^2}$$

This Hoeffding bound for each transition must not be greater than $\delta_P$. Therefore, for each possible transition (i.e. each possible and valid $P(s, a, s')$ combination),

$$Pr[|\frac{\#(s,a,s')}{\#(s,a)} - P(s, a, s')| \geq c] \leq \delta_P$$

$$2 \cdot e^{-2\#(s,a)c^2} \leq \delta_P$$

$$-2\#(s,a) \cdot c^2 \geq \ln(\delta_P/2)$$

$$c^2 \geq \frac{\ln(\delta_P/2)}{-2 \cdot \#(s,a)}$$

$$c \geq \sqrt{\frac{\ln(\delta_P/2)}{-2 \cdot \#(s,a)}}$$

Therefore, using a two-sided Hoeffding bound, the minimum confidence width of a transition $c$ is $c = \sqrt{\frac{\ln(\delta_P/2)}{-2\cdot\#(s,a)}}$, such that

$$Pr\left[\left|\frac{\#(s,a,s')}{\#(s,a)} - P(s, a, s')\right| \geq c\right] \leq \delta_P$$

## C. Collapsed MDP

**Definition C.1** (Super State). A super state is a single state that represents an MEC. Formally, if we have an MEC $E$ that is made of state set $S_E$, super state $ss$ is such that $ss = S \cap S_E$. $S_E$ is essentially treated as a single state. All staying transitions are self-loops, $(ss, (s, a), ss)$. All non-staying transitions are normal transitions with an altered state, action format. The non-staying transition $(s, a, s')$ becomes $(ss, (s, a), s')$. $s'$ can be a state or a super state. Note, a single state and the corresponding state-action pairs can be collapsed into a super state if the state and actions form an MEC. A more rigorous explanation of all cases for transition changes can be found in Appendix C.1.

**Definition C.2** (Collapsed MDP). A collapsed MDP $\mathcal{M}_C$ is an MDP where $\mathcal{M}_C = (S_C, A_C, S_{C,0}, P_C)$. $S_C$ contains all states not part of MECs and all collapsed MECs in the form of super states. $A_C$ includes all actions from states not in MECs and all actions from super states. An action from a super state is defined as a state-action pair. For example, if super state $ss$, has state $s_1$ with action $a$, the super state's action is $(s_1, a)$. $s_{C,0}$ is defined as the initial state or super state of the MDP, $s_{C,0} \in S_C$. $P_C$ is defined as the transition probabilities for the collapsed MDP. For $P_C(s, a, s')$, the states $s, s' \in S_C$ and action $a \in A_C$.

For BVI convergence, super states in the collapsed MDP are treated like normal states with one exception. If an action is

staying, its value bounds are $[0, 0]$. Formally,

$$L(s, a) = \begin{cases} 0 & \text{is\_staying\_action(a) and } G \cap E = \emptyset \\ 1 & \text{is\_staying\_action(a) and } G \cap E \neq \emptyset \\ \sum_{s':\#(s,a,s')>0} \hat{P}(s, a, s') \cdot L(s') & \text{else} \end{cases}$$

$$U(s, a) = \begin{cases} 0 & \text{is\_staying\_action(a) and } G \cap E = \emptyset \\ 1 & \text{is\_staying\_action(a) and } G \cap E \neq \emptyset \\ \left(\sum_{s':\#(s,a,s')>0} \hat{P}(s, a, s')\, U(s')\right) + \left(1 - \sum_{s':\#(s,a,s')>0} \hat{P}(s, a, s')\right) & \text{else} \end{cases}$$

### C.1. Collapse MDP Operation

Assuming all the MECs of an MDP have been correctly identified, the algorithm must deterministically collapse the MECs into super states; see Definition C.1. This turns MDP $\mathcal{M}$ into MDP $\mathcal{M}_C$, where $\mathcal{M} = (S, A, S_0, P)$ and $\mathcal{M}_C = (S_C, A_C, S_{C,0}, P_C)$. The following transition cases are defined for a collapsed MDP's behavior (Haddad & Monmege, 2017):

If a state $s$ takes action $a$ which transitions to another state $s'$, there is no difference with the topology or transition probabilities for $\mathcal{M}$ or $\mathcal{M}_C$.

If a state is a super state and it's transition leads to a normal state, then actions are rewritten to incorporate the state as well. If we look at state-action pair $(s, a)$ where $s$ is part of super state $ss$, transition $(s, a, s')$ becomes $(ss, (s, a), s')$.

If $s$ is a state and it takes action $a$ which has at least 1 transition that goes into super state $ss$, then all transitions that go into super state $ss$ from $s$ are combined into one single transition with the summed transition probability. For example, with $P(s, a, s')$ and $P(s, a, s'')$ where $s'$ and $s''$ are part of super state $ss$, $P_C(s, a, ss) = P(s, a, s') + P(s, a, s'')$.

Finally, if state $s$ is part of a super state $ss$ and has action $a$ that transitions to one or more states that are part of the super state $ss'$, the actions would incorporate the state as mentioned earlier, and the transitions would be merged into a single transition with the summed transition probability. For example, with $P(s, a, s')$ and $P(s, a, s'')$ where $s$ is part of super state $ss$, and $s'$ and $s''$ are part of super state $ss'$, $P_C(ss, (s, a), ss') = P(s, a, s') + P(s, a, s'')$.

### C.2. Collapse MDP Algorithm

The collapse MDP algorithm is the COLLAPSE_MECs(PM) algorithm as shown in Algorithm 1. To be $\delta_{EC}$ sure that the topology of the partial MDP is correct, we ensure each state-action pair has been sampled a total of $y$ times as mentioned in Appendix A.1. Then, Tarjan's algorithm is run recursive (every function call, all exit actions are removed). This results in a deterministic EC decomposition. Now that the ECs are known with $1 - \delta_{EC}$ confidence, the collapse MDP operation is run on each state, as mentioned in Appendix C.1

## D. Simulation Algorithms

Simulation algorithms are given in Algorithms 2 to 4.

Please note, the $\mu$ value influences the randomness of the simulation. $\mu$ should be set such that $\mu \in \mathbb{R}(0, 1]$ for the algorithm to work and for all the theoretical guarantees to hold. Intuitively, $\mu$ controls whether the explorative or exploitative nature of the algorithm dominates. A small value for $\mu$ will result in the best action heuristic being used more often while a larger $\mu$ value will result in a more explorative approach via. more random sampling.

## E. Proof of Theorem 3.1

### E.1. $\delta_k$ Split Guarantee Explanation for Theorem 3.1:

Putting everything together, there is at most (a) $\delta_{EC}$ confidence error of the collapsed MDP topology and EC detection being incorrect, (b) $\delta_{TP}$ confidence error of the transition probabilities being incorrect which influences $s_k$, and (c) $\delta_{N_k}$

---

**Algorithm 2** Simulate sampling from the true MDP

---

1: **function** SIMULATE(MDP $M$, target $G$)
2:     $X \leftarrow \emptyset$ {States visited during this simulation}
3:     $s \leftarrow s_0$
4:     **repeat**
5:         $X \leftarrow X \cup \{s\}$
6:         $a \leftarrow$ sampled uniformly with $1 - \mu$ probability from BEST_ACTION($s$) and with $\mu$ probability from all actions
7:         $s' \leftarrow$ sampled according to $P(s, a)$
8:         INCREMENT $\#(s, a, s')$
9:         $s \leftarrow s'$
10:     **until** $s \in G$ **or** LOOPING($X, s$)
11:     **return** $X \cup \{s\}$ {Include last state in returned path}
12: **end function**

---

**Algorithm 3** Looping - Check whether we are probably looping and should stop the simulation

---

1: **function** LOOPING(State set $X$, State $s$) {$X$ = visited states, $s$ = current state}
2:     **if** $s \notin X$ **then**
3:         **return false** {No loop possible yet}
4:     **end if**
5:     **return** $\exists C \subseteq X : C$ is an EC in partial model $\land s \in C \land \delta_C$-SURE-EC($C$)
6: **end function**

---

confidence error that $N_k$ simulations does not sample every relevant transition at least $s_k$ times. This is the start to finish of the proof. It was defined in Appendix A.1 that $\delta_k = \delta_{TP} + \delta_{EC} + \delta_{N_k}$. This means that there exists a $K_{PAC}$ s.t. for every $k \geq K_{PAC}$, there exists an $N_k$ s.t. with confidence at least $(1 - \delta_k)$, $U(s_{C,0}) - L(s_{C,0}) \leq \varepsilon_k$. Note, the lower and upper bound found are actually for $s_{C,0}$ instead of $s_0$, since BVI is run on the collapsed partial MDP. For simplicity, this is left our of the proof sketch. However, it is seen in the rigorous proof of the theorem.

### E.2. Relevant State-Actions/Transitions

A *relevant* action is an action unique to a state, $s$. It is such that $a \in Av(s)$ where $a \in best\_action(s)$ and state $s$ has reachability from $s_0 > 0$ if the algorithm runs a simulation using the guiding heuristic in Algorithm 2. Note that the guiding heuristic utilizes $U$ from stage $k - 1$ of the algorithm. Additionally, a relevant transition is any transition seen from a relevant action.

### E.3. Existence and Calculation of $s_k$

**Lemma E.1.** *Let $\hat{M} = (S, A, \hat{P}, s_0)$ be our partial MDP and $\hat{M}_C = (S_C, A_C, \hat{P}_C, s_{C,0})$ be $\hat{M}$ collapsed as mentioned in Algorithm 1. If $k \geq K_{PAC}$, then after running BVI on $\hat{M}_C$, if $Pr[U(s_{C,0}) - L(s_{C,0}) \leq \varepsilon_k] \geq 1 - \delta_{TP}$, then $Pr[U(s_0) - L(s_0) \leq \varepsilon_k] \geq 1 - (\delta_{TP} + \delta_{EC})$*

*Proof.* The algorithm runs BVI on the collapsed MDP, which is derived from the discovered partial MDP, as mentioned in Section 3.2. Therefore, we can only provide guarantees on $U(s_0) - L(s_0) \leq \varepsilon_k$ if we provide guarantees on the collapsed partial model.

Note that the algorithm is $1 - \delta_{EC}$ sure that the topology of the discovered MDP is correct, given that each state-action pair has been sampled at least $y$ times as mentioned in Appendix A.1. EC decomposition is deterministic and based on MDP topology. The algorithm has collapsed the discovered partial MDP into $\hat{M}_C$, and with high confidence, $\hat{M}_C$ has the same topology as the ground truth collapsed MDP, $\mathcal{M}_C$, where $\mathcal{M}_C = (S_C, A_C, s_{C,0}, P_C)$ (for all reachable states). This means the collapsed partial MDP is correct w.r.t. the collapsed version of the true MDP with $\delta_{EC}$ error.

$\hat{M}$ and $\hat{M}_C$ will have the same optimal policy. The only difference between these two MDPs is the set of states in ECs. The optimal policy for such states would be to deterministically take actions that stay in the EC until we visit the state where $best\_exit\_action(EC)$ exists. Once we visit that state, the best exit action is taken. This process is described in Section 3.2.

---

**Algorithm 4** Checks whether $C$ is an end-component with $\delta_C$ confidence

---
1: **function** $\delta_C\_\text{SURE\_EC}$(State set $C$, min probability $p_k$)
2:     $required\_samples \leftarrow \dfrac{\ln(\delta_C)}{\ln(1 - p_k)}$
3:     $B \leftarrow \{(s, a) \mid s \in C \; \wedge \; \neg(s, a) \text{ exits } C\}$ {Staying state-action pairs}
4:     **return** $\displaystyle\bigwedge_{(s,a) \in B} \left(\#(s, a) > required\_samples\right)$
5: **end function**

---

For the collapsed MDP, once we land in a super state, we simply choose the best exit action since all the EC states are now a single state. Thus, we can optimally preserve and translate the policy of $\hat{\mathcal{M}}_C$ to a policy for $\hat{\mathcal{M}}$ with the sample reachability value. If we find a policy $\pi_C$ that is an $\varepsilon_k$-optimal policy for $\mathcal{M}_C$, then with probability $1 - \delta_{EC}$, we can translate $\pi_C$ into policy $\pi$ which is $\varepsilon_k$-optimal for $\mathcal{M}$.

A policy on $\mathcal{M}_C$ is optimal if and only if BVI converges such that $U(s_{C,0}) - L(s_{C,0}) \leq \varepsilon_k$. Moreover, we know the estimated transition probabilities have at most $\delta_{TP}$ error. Therefore, we say, if $Pr[U(s_{C,0}) - L(s_{C,0}) \leq \varepsilon_k] \geq 1 - \delta_{TP}$, then $Pr[U(s_0) - L(s_0) \leq \varepsilon_k] \geq 1 - (\delta_{TP} + \delta_{EC})$. $\qquad\square$

**Lemma E.2.** *In Algorithm 1, there exists a $K_{\text{PAC}} \in \mathbb{N}$ s.t. for all $k \geq K_{\text{PAC}}$, if all relevant transitions are sampled at least $s_k$ times, then*
$$\Pr[\pi_k \in \Pi_{\text{opt}}^{\varepsilon_k}] \geq 1 - (\delta_{TP} + \delta_{EC}).$$

*Proof.* Note, this proof of calculating $s_k$ was inspired from (Ashok et al., 2019).

Let the integer $K_{\text{PAC}}$ be such that $p_{K_{\text{PAC}}} = \dfrac{1}{2^{K_{\text{PAC}}}} \leq p_{min}$ where $p_{min}$ is the minimum non-zero probability in the ground truth MDP $\mathcal{M}$. Note that even though the value $p_{min}$ is unknown, such a $K_{\text{PAC}}$ will certainly exist.

A policy $\pi$ is $\varepsilon_k$-optimal if $U(s_0) - L(s_0) \leq \varepsilon_k$. This is because $L(s_0) \leq V(s_0) \leq U(s_0)$. Any value taken within the range of the lower and upper bound of the initial state is guaranteed to be within $\varepsilon_k$ of $V(s_0)$.

We aim to prove that there exists a $K_{PAC}$ s.t. for all $k \geq K_{PAC}$, there exists an $N_k$ s.t. $Pr[U(s_0) - L(s_0) \leq \varepsilon_k] \geq 1 - \delta_k$.

Since Algorithm 1 collapses the partial MDP to handle ECs, we can use Lemma E.1 and say that by proving BVI finds an $\varepsilon_k$-optimal policy for the collapsed MDP, it has found an $\varepsilon_k$-optimal policy for the uncollapsed MDP as well, with $\delta_{EC}$ more probabilistic error. Formally, if $k \geq K_{PAC}$, then after running BVI on $\hat{\mathcal{M}}_C$, if $Pr[U(s_{C,0}) - L(s_{C,0}) \leq \varepsilon_k] \geq 1 - \delta_{TP}$, then $Pr[U(s_0) - L(s_0) \leq \varepsilon_k] \geq 1 - (\delta_{TP} + \delta_{EC})$.

Now, it will be proven that for the collapsed discovered MDP, BVI converges. Proving BVI convergence for the collapsed discovered MDP proves the convergence of the uncollapsed MDP since no transitions are lost, and collapsing the MDP is actually the correct way of deflating $U$. Therefore, it must be proven that $Pr[U(s_{C,0}) - L(s_{C,0}) \leq \varepsilon_k] \geq 1 - \delta_{TP}$ where $s_{C,0}$ is the initial state of the collapsed discovered MDP. Because we are at least $1 - \delta_{EC}$ confident that the collapsed discovered MDP's topology is equal to that of $\mathcal{M}_C$, it can be said that the initial states for both collapsed MDPs are $s_{C,0}$ with at least $1 - \delta_{EC}$ confidence.

We can prove this lemma if the lower bound of the collapsed discovered MDP's initial state can be bounded such that $V(s_{C,0}) - L(s_{C,0}) \leq \frac{\varepsilon_k}{2}$ since this can be done for the upper bound, $U(s_0)$, as well. The logic remains the same.

$$V(s_{C,0}) - L(s_{C,0}) \leq \frac{\varepsilon_k}{2}$$
$$U(s_{C,0}) - V(s_{C,0}) \leq \frac{\varepsilon_k}{2}$$
$$U(s_{C,0}) - L(s_{C,0}) \leq \varepsilon_k$$

This proof sums error along MDP paths. Thus, the proof requires the MDP to be acyclical.

Let $\mathcal{M}_r$ be the unrolled version of MDP $\mathcal{M}_C$ with a step counter that allows for a maximum of $r$ steps. Formally, $\mathcal{M}_r$ has the state space $S_C \times [r]$, where $[r]$ is the set of all natural numbers in the range $[0, r]$. Actions and transitions work the same,

except they increment the step counter. Transition $(s, a, s')$ becomes $((s, r), a, (s', r + 1))$. The step counter makes $\mathcal{M}_r$ acyclical since it is not possible to go back to a smaller $r$. Additionally, the depth of the MDP is $r$ steps. This means that any transition going to a state where the value of the step counter is greater than $r$ leads to a sink state, $ss$ with $V(ss) = 0$.

Let $V(\mathcal{M}_C)$ denote the true value from the initial state of the true, collapsed MDP. Formally, $V(\mathcal{M}_C) = V(s_{C,0})$. For an unrolled MDP, the reachability objective is satisfied when a target/goal state is reached in the state component of the MDP. It can be said that $V(\mathcal{M}_r) \leq V(\mathcal{M}_C)$. This is because the reachability for $\mathcal{M}_r$ is an r-horizon task, i.e. a task that can be done with at most $r$ steps. However, $V(\mathcal{M}_C)$ is an infinite horizon task as there is no limit to the number of steps the agent can take. Therefore, the paths that reach the goal for $\mathcal{M}_C$ are the paths that reach the goal for $\mathcal{M}_r$ plus the paths of length $(r + c)$ where $c \in \mathbb{N}^+$. The paths shorter than $r$ step have the same reachability probability for both $\mathcal{M}_C$ and $\mathcal{M}_r$. Hence, $V(\mathcal{M}_r) \leq V(\mathcal{M}_C)$.

It can also be shown that $\lim_{r \to \infty} V(\mathcal{M}_r) = V(\mathcal{M}_C)$. As $r$ increases, the number of paths that satisfy the reachability objective either stays equal or increases. As $r \to \infty$, the reachability objective turns into one for an infinite-horizon task, meaning all possible paths that satisfy the reachability objective will be accounted for. Thus, $\lim_{r \to \infty} V(\mathcal{M}_r) = V(\mathcal{M}_C)$.

Let $V(\hat{\mathcal{M}}_r)$ be the value of $s_{C,0}$ in $\mathcal{M}_r$ if the estimated transition probabilities, $\hat{P}_C$, were used instead of $P_C$. All remaining probabilities will be directed towards a sink state of value 0. $V(\hat{\mathcal{M}}_C)$ is the limit of BVI for the partial MDP constructed according to (Ashok et al., 2019). By this definition $V(\hat{\mathcal{M}}_C)$ is the limit of the best lower bound, $L(s_{C,0})$, which BVI can obtain using $\hat{P}$. Let the number of BVI iterations be at least $r$ (which can be assumed since the number of BVI iterations increases more than linearly with $k$). This means that the lower bound calculated from BVI is $L(s_{C,0}) \geq V(\hat{\mathcal{M}}_r)$, given the number of BVI iterations is at least $r$ according to (Ashok et al., 2019).

The objective is to prove that with $1 - \delta_k$ confidence,

$$V(s_{C,0}) - L(s_{C,0}) \leq \frac{\varepsilon_k}{2}$$

Assume an $r$ has been chosen such that

$$V(\mathcal{M}_C) - V(\mathcal{M}_r) \leq \frac{\varepsilon_k}{4}$$

Such an $r$ is guaranteed to exist as $\lim_{r \to \infty} V(\mathcal{M}_r) = V(\mathcal{M}_C)$. The explanation above states that $L(s_{C,0}) \geq V(\hat{\mathcal{M}}_r)$. Therefore, the condition can be rewritten.

$$
\begin{aligned}
V(s_{C,0}) - L(s_{C,0}) &\leq \frac{\varepsilon_k}{2} \\
V(s_{C,0}) - L(s_{C,0}) &\leq V(s_{C,0}) - V(\hat{\mathcal{M}}_r) \\
&= V(\mathcal{M}_C) - V(\hat{\mathcal{M}}_r) \\
&= V(\mathcal{M}_C) - V(\mathcal{M}_r) \\
&\quad + V(\mathcal{M}_r) - V(\hat{\mathcal{M}}_r) \\
&\leq \frac{\varepsilon_k}{4} + V(\mathcal{M}_r) - V(\hat{\mathcal{M}}_r)
\end{aligned}
$$

It is shown that

$$
\begin{aligned}
\frac{\varepsilon_k}{4} + V(\mathcal{M}_r) - V(\hat{\mathcal{M}}_r) &\leq \frac{\varepsilon_k}{2} \\
V(\mathcal{M}_r) - V(\hat{\mathcal{M}}_r) &\leq \frac{\varepsilon_k}{4}
\end{aligned}
$$

Therefore, to prove convergence, it must be proved that $V(\mathcal{M}_r) - V(\hat{\mathcal{M}}_r) \leq \frac{\varepsilon_k}{4}$. In other words, the error of using $\hat{P}_C$ instead of $P_C$ should be bounded by $\frac{\varepsilon_k}{4}$.

Given Lemma E.3, we see that for all $i \in \mathbb{Z}$ where $i \geq 0$, $P : V_r(s) - \hat{V}_r(s) \leq 2c \cdot \left(\frac{1}{p_k}\right)^i \cdot i$, where $V_r(s)$ and $\hat{V}_r(s)$ are the values of a state $s \in S_C \times [r]$ for MDPs $\mathcal{M}_r$ and $\hat{\mathcal{M}}_r$ respectively. Note, $i$ denotes the longest path length from state $s$ to an accepting state or sink state.

The longest path in the unrolled MDP, $\mathcal{M}_r$ is $r$ steps since each transition in $\mathcal{M}_r$ increments the step counter by 1, and after $r$ steps, we reach a terminal state. Therefore, for any state $s \in S$,

$$V_r(s) - \hat{V}_r(s) \leq 2c \cdot \left(\frac{1}{p_k}\right)^r \cdot r$$

Now, the confidence width $c$ must be found such that the error of using $\hat{P}_C$ instead of $P_C$ is less than $\frac{\varepsilon_k}{4}$. To do this,

$$V_r(s) - \hat{V}_r(s) \leq 2c \cdot \left(\frac{1}{p_k}\right)^r \cdot r$$

$$2c \cdot \left(\frac{1}{p_k}\right)^r \cdot r \leq \frac{\varepsilon_k}{4}$$

The confidence width, $c$, is calculated using the two-sided Hoeffding bound as mentioned in Section 3.2

$$c = \sqrt{\frac{\ln(\delta_P/2)}{-2 \cdot \#(s,a)}}$$

Substituting results in

$$2 \cdot \sqrt{\frac{\ln(\delta_P/2)}{-2 \cdot \#(s,a)}} \cdot \left(\frac{1}{p_k}\right)^r \cdot r \leq \frac{\varepsilon_k}{4}$$

where $\#(s,a)$ is the number of samples require for each state-action pair. We define $s_k$ (mentioned earlier in the proof) as $s_k = \#(s,a)$. Rewriting,

$$s_k \geq \frac{32 \cdot ln(2/\delta_P) \cdot r^2}{\varepsilon_k^2 \cdot p_k^{2 \cdot r}}$$

If every relevant transitions has been sampled at least $s_k$ times, the necessary confidence width would be achieved for the algorithm to converge to a maximum error of $\varepsilon_k$. This is because, sampling every relevant transition at least $s_k$ times means

$$V(\mathcal{M}_r) - V(\hat{\mathcal{M}}_r) \leq \frac{\varepsilon_k}{4} .$$

Since it is assumed that

$$V(\mathcal{M}_C) - V(\mathcal{M}_r) \leq \frac{\varepsilon_k}{4}$$

it can be said that

$$V(s_0) - L(s_0) \leq \frac{\varepsilon_k}{2}$$

The same argument can be made for the upper bound. Henceforth, if each relevant state-action (or each relevant transition if we take an upper bound) has been seen at least $s_k$ times, $U(s_{C,0}) - L(s_{C,0}) \leq \varepsilon_k$.

Note, the induction argument is satisfied for relevant actions in a state because when induction is done backwards from a target or sink state all the way to $s_0$, we are only looking at actions that we would select (as mentioned by the definition and choice of $a_1$ and $a_2$ in the setup of the induction). We are creating an optimal policy. The optimal policy is created by sampling the best actions. The optimal policy will be only the best actions from each state. Therefore, the set of all relevant actions is equal to the set of all actual actions that could be taken. Hence, if we sample each relevant transition $s_k$ times and have the error bound for seeing each relevant transition, it will mean the error is bound for all actions used in the policy. Thus we need to make sure the error bounded by each state-action pair seen in this optimal policy is withing $\varepsilon_k$. As a result, we only care about relevant transitions.

To ensure there are no issues with the MDP's topology, each transition (not just relevant transitions) must be sampled at least $y$ times where $y \geq \frac{\ln(\delta_C)}{\ln(1-p_k)}$ as mentioned in Appendix A.1. Since $y << s_k$, we can say that *all* transitions must be sampled at least $s_k$ times for us to guarantee that $U(s_{C,0}) - L(s_{C,0}) \leq \varepsilon_k$ with high confidence.

Additionally, by requiring each transition to be seen at least $s_k$ times, we can account for the rare case when the *relevant* actions seen are not the *true* relevant actions. By not restricting the sampling requirement to only relevant actions, we ensure the algorithm adjusts and corrects itself in the future, regardless of previous stage outputs.

In this proof, the only probabilistic errors possible to prove that $U(s_{C,0}) - L(s_{C,0}) \leq \varepsilon_k$ with $s_k$ samples were from collapsing ECs and estimating transition probabilities. These two parts have confidence errors $\delta_{EC}$ and $\delta_{TP}$ as defined in Appendix A.1. Therefore, we see that if each relevant transition is sampled at least $s_k$ times, $\Pr[\pi_k \in \Pi^{\varepsilon_k}_{\mathrm{opt}}] \geq 1 - (\delta_{TP} + \delta_{EC})$. $\qquad\square$

**Lemma E.3.** *For all $i \in \mathbb{Z}$ where $i \geq 0$, $V_r(s) - \hat{V}_r(s) \leq 2c \cdot \left(\frac{1}{p_k}\right)^i \cdot i$. We define $V_r(s)$ and $\hat{V}_r(s)$ as the value of a state $s \in S_C \times [r]$ for MDPs $\mathcal{M}_r$ and $\hat{\mathcal{M}}_r$ respectively. Note, $i$ denotes the longest path length from state $s$ to an accepting state or sink state.*

*Proof.* The following induction argument proves that this inequality will be satisfied. Intuitively, we bound the approximation error in estimating the value of a state. Let $V_r(s)$ and $\hat{V}_r(s)$ be the value of a state $s \in S_C \times [r]$ for MDPs $\mathcal{M}_r$ and $\hat{\mathcal{M}}_r$ respectively. The proposition is as follows. For every state $s \in S_C \times [r]$,

$$P : V_r(s) - \hat{V}_r(s) \leq 2c \cdot \left(\frac{1}{p_k}\right)^i \cdot i$$

where $i \geq 0$, and $i$ denotes the longest path length from state $s$ to an accepting state or sink state. Because $\mathcal{M}_r$ is acyclic and unrolled to a maximum of $r$ steps, $i \leq r$. Therefore, by proving, this inequality for all $s \in S_C \times [r]$, it inherently proves that

$$V_r(s_{C,0}) - \hat{V}_r(s_{C,0}) \leq 2c \cdot \left(\frac{1}{p_k}\right)^r \cdot r$$

$$V(\mathcal{M}_r) - V(\hat{\mathcal{M}}_r) \leq 2c \cdot \left(\frac{1}{p_k}\right)^r \cdot r$$

meaning, $V(\mathcal{M}_r) - V(\hat{\mathcal{M}}_r)$ is bounded, and with the right confidence width $c$, can be less than $\frac{\varepsilon_k}{4}$.

Continuing the induction:

Base case: let $i = 0$. The subset of states with longest path of 0 steps are sink states and goal states. For all these states,

$$V_r(s) - \hat{V}_r(s) \leq 2c \cdot \left(\frac{1}{p_k}\right)^0 \cdot 0$$

$$V_r(s) - \hat{V}_r(s) \leq 0$$

This is accuracy as all goal states and sink states have values $1$ and $0$ respectively for $\mathcal{M}_r$ and $\hat{\mathcal{M}}_r$. The value difference is $0$ because no transitions occur on these paths, meaning $P_C$ and $\hat{P}_C$ have note been used.

Inductive Hypothesis: for a state $s \in S_C \times [r]$ that reaches a goal or sink state with longest path of length at most $i$ steps,

$$V_r(s) - \hat{V}_r(s) \leq 2c \cdot \left(\frac{1}{p_k}\right)^i \cdot i$$

Induction Step: for any state $s \in S_C \times [r]$ that reaches a goal or a sink state with longest path of length at most $i + 1$, let $a_1$ be the best possible (highest value) action from $s$ that maximizes $V_r(s)$. Let $a_2$ be the best possible action from $s$ that maximizes $\hat{V}_r(s)$. More formally, $a_1 \in argmax_{a_1 \in Av(s)} V_r(s, a_1)$ and $a_2 \in argmax_{a_2 \in Av(s)} \hat{V}_r(s, a_2)$, where $V_r(s, a)$ and $\hat{V}_r(s, a)$ are the values of state-action pairs in $\mathcal{M}_r$ and $\hat{\mathcal{M}}_r$. From this,

$$\begin{aligned}
V_r(s) - \hat{V}_r(s) &= V_r(s, a_1) - \hat{V}_r(s, a_2) \\
&\leq V_r(s, a_1) - \hat{V}_r(s, a_1) \\
&= \sum_{s'} \left[ P(s, a_1, s') V_r(s') - \hat{P}(s, a_1, s') \hat{V}_r(s') \right]
\end{aligned}$$

Because there are at most $\frac{1}{p_k}$ transitions per state-action pair, $s''$ can be defined as the state that maximizes $P(s, a_1, s')V_r(s') - \hat{P}(s, a_1, s')\hat{V}_r(s')$. Thus, the expression can be rewritten as

$$\leq \frac{1}{p_k} \left[ P(s, a_1, s'')V_r(s'') - \hat{P}(s, a_1, s'')\hat{V}_r(s'') \right]$$

Now, if it is shown that

$$P(s, a_1, s'')V_r(s'') - \hat{P}(s, a_1, s'')\hat{V}_r(s'') \leq 2c \left( \frac{1}{p_k} \right)^i (i+1)$$

the induction will be complete as

$$\leq \frac{1}{p_k} \left[ P(s, a_1, s'')V_r(s'') - \hat{P}(s, a_1, s'')\hat{V}_r(s'') \right]$$

$$\leq \frac{1}{p_k} \cdot 2c \left( \frac{1}{p_k} \right)^i (i+1)$$

$$= 2c \left( \frac{1}{p_k} \right)^{i+1} (i+1)$$

Note that $s''$ is a successor of $s$. Because the longest path length from $s$ to a goal or sink state is at most $i+1$, the longest path from $s''$ to a goal or sink state must be at most $i$. Therefore, $V_r(s'') - \hat{V}_r(s'')$ can be used as the inductive hypothesis.

Before continuing the proof of the induction step, it is known that with the two-sided Hoeffding bound on the transition probabilities, the algorithm is $1 - \delta_{TP}$ confident that for all transitions,

$$\left| P(s, a, s') - \frac{\#(s, a, s')}{\#(s, a)} \right| \geq c$$

Considering that

$$\hat{P}(s, a, s') = \frac{\#(s, a, s')}{\#(s, a)} - c$$

with at most $\delta_{TP}$ confidence error, it can be assumed that for all $\hat{P}(s, a, s')$,

$$P(s, a, s') - 2c \leq \hat{P}(s, a, s') \leq P(s, a, s')$$

Now, the induction is continued by proving the following:

$$= P(s, a_1, s'')V_r(s'') - \hat{P}(s, a_1, s'')\hat{V}_r(s'')$$

$$\leq P(s, a_1, s'')V_r(s'') - (P(s, a_1, s'') - 2c)\hat{V}_r(s'')$$

$$= P(s, a_1, s'') \cdot (V_r(s'') - \hat{V}_r(s'')) + 2c \cdot \hat{V}_r(s'')$$

$$\leq P(s, a_1, s'') \cdot 2c \cdot \left( \frac{1}{p_k} \right)^i \cdot i + 2c \cdot \hat{V}_r(s'')$$

$$= 2c \cdot \left[ \left( \frac{1}{p_k} \right)^i \cdot i \cdot P(s, a_1, s'') + \hat{V}_r(s'') \right]$$

$$\leq 2c \cdot \left[ \left( \frac{1}{p_k} \right)^i \cdot i \cdot 1 + \hat{V}_r(s'') \right]$$

$$\leq 2c \cdot \left[ \left( \frac{1}{p_k} \right)^i \cdot i \cdot 1 + 1 \right]$$

$$\leq 2c \cdot \left( \frac{1}{p_k} \right)^i \cdot (i+1)$$

Therefore, it is shown that

$$\leq \frac{1}{p_k} \left[ P(s, a_1, s'') V_r(s'') - \hat{P}(s, a_1, s'') \hat{V}_r(s'') \right]$$

$$= \frac{1}{p_k} \cdot 2c \left( \frac{1}{p_k} \right)^i (i+1)$$

$$= 2c \left( \frac{1}{p_k} \right)^{i+1} (i+1)$$

Hence, it has been proven that for all $s \in S_C \times [r]$ and for all $i \geq 0$

$$V_r(s) - \hat{V}_r(s) \leq 2c \cdot \left( \frac{1}{p_k} \right)^i \cdot i$$

$\square$

### E.4. Calculating the Probability of Seeing the Least Likely Relevant Transition

**Lemma E.4.** *The lower bound for the probability of the least likely transition occurring in a single simulation is*

$$p \geq \left( \frac{\mu \cdot p_k}{|A|} \right)^{|S|}$$

.

*Proof.* $p$ is defined as the probability that the least likely transition occurs in a single simulation of the algorithm. Because of $\mu$ (defined in Appendix D), there is a non-zero chance that the algorithm lands on state $s$ of any (relevant and/or non-relevant) action, and there is a non-zero chance that the action $a$ is sampled. Therefore, for every action, there is a non-zero chance that the action is seen. Because a transition is any seen transition from an action, it is implied that transitions also have a non-zero probability of being seen during a simulation.

Now we know that $p > 0$. To calculate the minimum value bound of $p$, we consider the least likely transition. Such a transition would require the algorithm to go through at most $|S|$ states during the simulation to get to state $s$. State $s$ is where this action can be run. In the simulation, going from one state to another requires either sampling the best action or random sampling, based on $\mu$ as mentioned in Appendix D. In the worst case, the action is not one of the best actions to take, and we need the simulation algorithm to randomly sample the action. there will be at most $|A|$ actions from the current state; only one of those actions has a transition to the next desired state, and that transition is with minimum probability $p_k$. Therefore, going from one state to another in the simulation will have a minimum probability of $\frac{\mu \cdot p_k}{|A|}$. Because we need to traverse through at most $|S|$ states, we will take at most $|S| - 1$ steps. Thus, we can get to the state where the least likely transition can occur with probability $\geq \left( \frac{\mu \cdot p_k}{|A|} \right)^{|S|-1}$. Finally, the probability of seeing the relevant transition is at least $\frac{\mu \cdot p_k}{|A|}$.

Therefore, the minimum probability of seeing the least likely relevant transition is

$$p \geq \left( \frac{\mu \cdot p_k}{|A|} \right)^{|S|}.$$

$\square$

### E.5. Existence of $N_k$

**Lemma E.5.** *Given that each transition must be seen at least $s_k$ times, and given that the least likely transition is seen with probability $p \geq \left( \frac{\mu \cdot p_k}{|A|} \right)^{|S|}$ in a single simulation, there exists an $N_k$ s.t. the probability of not seeing each transition at least $s_k$ times is bounded by at most $\delta_{N_k}$ as defined in Appendix A.1.*

*Proof.* We assume that $p = \left( \frac{\mu \cdot p_k}{|A|} \right)^{|S|}$ is the minimum probability with which a relevant transition can be done. Let $X$ be a random variable that counts the number of times the least likely transition has occurred. $X$ follows a Binomial Distribution with parameters $(N_k, p)$, meaning $X \sim B(N_k, p)$. The cumulative distribution function (CDF) of $X$ is $F(X \leq s_k - 1) = \sum_{i=0}^{s_k - 1} \binom{N_k}{i} \cdot p^i \cdot (1-p)^{N_k - i}$. This CDF gives the probability of $N_k$ simulations not choosing the least likely relevant transition at least $s_k$ times. It would be ideal for $F(X \geq s_k) = 1$. Unfortunately, this is impossible as there will always be some probability that fewer than $s_k$ samples are seen: $F(X < s_k) > 0$. However, the algorithm can choose a large $N_k$ such that $F(X < s_k) \leq \delta_n$ (which would be possible with a large enough $N_k$, i.e. there exists an $N_k$ s.t. $F(X < s_k) \leq \delta_n$ since $\lim_{N_k \to \infty} F(X < s_k) = 0$). Note, $\delta_n$ is the upper bound for the error that a relevant transition was not seen $s_k$ times. $\delta_n$ has been defined in Appendix A.1 as $\delta_n = \frac{\delta_{N_k} \cdot p_k}{|SA|}$, where $|SA|$ is the number of state-action pairs discovered. $\delta_{N_k}$ is essentially partitioned into several $\delta_n$ errors that are consumed by each relevant transtion. Since the number of relevant transitions is no greater than the number of possible transitions, $\frac{|SA|}{p_k}$, the confidence error of not seeing a single relevant transition at least $s_k$ times is bounded by $\delta_{N_k}$. Hence, with $\delta_{N_k}$ confidence error, it can be said that given $s_k$ is the minimum number of times to see the least likely relevant transition, running $N_k$ simulations will run each relevant transition at least $s_k$ times. $\qquad\square$

## E.6. Entire Proof of Theorem 3.1

**Theorem 3.1** In Algorithm 1, there exists a $K_{\mathsf{PAC}} \in \mathbb{N}$ s.t. for all $k \geq K_{\mathsf{PAC}}$ there exists a number of simulations $N_k$ s.t.

$$\Pr[\pi_k \in \Pi_{\mathsf{opt}}^{\varepsilon_k}] \geq 1 - \delta_k.$$

I.e. policy $\pi_k$ is $\varepsilon_k$-optimal in MDP $\mathcal{M}$ with confidence $1 - \delta_k$.

*Proof.* By Lemma E.2, in Algorithm 1, there exists a $K_{\mathsf{PAC}} \in \mathbb{N}$ s.t. for all $k \geq K_{\mathsf{PAC}}$, all transitions must be sampled at least $s_k$ times to guarantee $\Pr[\pi_k \in \Pi_{\mathsf{opt}}^{\varepsilon_k}] \geq 1 - (\delta_{TP} + \delta_{EC})$.

By Lemma E.4, there exists a minimum probability for the least likely transition, $p$. Lemma E.4 also states that $p \geq \left( \frac{\mu \cdot p_k}{|A|} \right)^{|S|}$.

We now can assume that each transition must be seen at least $s_k$ times and that the least likely transition is seen with probability $p \geq \left( \frac{\mu \cdot p_k}{|A|} \right)^{|S|}$. Therefore, by Lemma E.5, there exists an $N_k$ s.t. the probability of not seeing each relevant transition at least $s_k$ times is bounded by at most $\delta_{N_k}$.

The above statement proves that there exists an $N_k$ for an $s_k$, given $\delta_{N_k}$. This, along with Lemma 1, indicates that in Algorithm 1, there exists a $K_{\mathsf{PAC}} \in \mathbb{N}$ s.t. for all $k \geq K_{\mathsf{PAC}}$ there exists a number of simulations $N_k$ s.t.

$$\Pr[\pi_k \in \Pi_{\mathsf{opt}}^{\varepsilon_k}] \geq 1 - (\delta_{TP} + \delta_{EC}) - \delta_{N_k}.$$

Because $\delta_k = \delta_{EC} + \delta_{TP} + \delta_{N_k}$ (as shown in Appendix A.1), we hence prove that in Algorithm 1, there exists a $K_{\mathsf{PAC}} \in \mathbb{N}$ s.t. for all $k \geq K_{\mathsf{PAC}}$ there exists a number of simulations $N_k$ s.t.

$$\Pr[\pi_k \in \Pi_{\mathsf{opt}}^{\varepsilon_k}] \geq 1 - (\delta_k).$$

I.e. policy $\pi_k$ is $\varepsilon_k$-optimal in MDP $\mathcal{M}$ with confidence $1 - \delta_k$.

Note, it must be true that $k \geq K_{PAC}$. Otherwise, our estimate of the minimum transition probability, $p_k$, would be greater than the true $p_{min}$. This would prevent the assumptions in the proof from being correct. $\qquad\square$

## F. Technical details for bound on $\varepsilon_{\mathsf{diff}}$

In this section, we provide the missing proofs and technical details from Section 4.

**Matrix expression.** Here, we motivate the formulation of Equation (1). In a known environment $\mathcal{M}$, given a memoryless deterministic policy $\pi$, we get a value $v_s$ for a state $s$. In reachability for all transient states $s \in S \setminus G$, we have

$$v_s = \sum_{s' \in S} P(s, \pi(s), s') v_{s'}$$

and for $s \in G$, we have $v_s = w_s$, where $w_s$ is the general reward (usually $1$ or $0$). To argue about the exact values, we expand the equation and express the values of choosing state-action pairs

$$v_{s,a} = \sum_{s' \in S} P(s, a, s')v_{s'},$$

then we express the values of states under a deterministic policy as

$$v_s = v_{s,\pi(s)}.$$

Observe that all policies can be encoded in such a way. Two policies only differ in the choice of $v_s$.

Note that the solution to Equation (1) might not be unique; the value of reachability is the smallest solution to the equation. However, we suppose the MDP $\mathcal{M}$ is *halting*: all policies eventually reach some $s \in G$. Halting MDPs have only a unique solution to the equation $v = v\mathbf{A} + \mathbf{r}$.

**Transition complexity.**   Here, we describe the transition complexity in more detail. The transition complexity $D_i$ of a matrix row is a number such that multiplying all entries in row $i$ by $D_i$ gives integers. For $s \in S$ in the row $\mathbf{A}_{s,\cdot}$, we have $D_s = 1$, since all entries are either $0$ or $1$. For $s \in S$ and $a \in A$, for the row $\mathbf{A}_{(s,a),\cdot}$, we express $P_a(s, s')$ for $a \in A$ and $s, s' \in S$ as a fraction

$$P(s, a, s') = \frac{p_{(s,a),s'}}{q_{(s,a),s'}}$$

where $p_{(s,a),s'}, q_{(s,a),s'} \in \mathbb{Z}$. We define

$$D_{s,a} = \mathrm{lcm}_{s' \in S}(q_{(s,a),s'}),$$

where lcm is the lowest common multiple of inputs. Then, the transition complexity of the whole matrix is defined as the maximum over all rows. In our case, it is

$$D = \max_{(s,a) \in S \times A} (D_{s,a}).$$

*Proof of Theorem 4.1.*  We rewrite the equation $v = v\mathbf{A} + \mathbf{r}$ as

$$v(Id - \mathbf{A}) = \mathbf{r},$$

where $Id$ is an identity matrix. We define a diagonal matrix $F$ indexed by $S$ and $S \times A$ such that $F_{s,s} = 1$ and $F_{(s,a),(s,a)} = D_{s,a}$. Then, in the matrix $F(Id - \mathbf{A})$, the corresponding row with index $(s,a)$ is multiplied by its transition complexity $D_{s,a}$. That means the entries of the matrix $F(Id - \mathbf{A})$ are all integers.

Now, we argue about the solution of

$$vF(Id - \mathbf{A}) = F\mathbf{r}.$$

To solve the matrix equation, we can use Cramer's rule to get an explicit expression of the solution. In our situation, Cramer's rule states that

$$v_i = \frac{\det(B_i)}{\det(F(Id - \mathbf{A}))},$$

where $B_i$ be the matrix created from $F(Id - \mathbf{A})$ by replacing its column with index $i$ by the vector $F\mathbf{r}$. Moreover, all entries of the matrix $B_i$ are integers since $\mathbf{r}$ is an integer vector, $F$ is an integer diagonal matrix, and $F(Id - \mathbf{A})$ is an integer matrix. Observe that the determinant of an integer matrix is an integer since the determinant only sums products of different permutations.

Since $\det(B_i)$ is always an integer, we bound the value of $\det(F(Id - \mathbf{A}))$. From multiplicativity of determinant, we have $\det(F(Id - \mathbf{A})) = \det(F) \cdot \det(Id - \mathbf{A})$.

We bound $\det(F)$, but the determinant of a diagonal matrix is only the product of all entries. Therefore, we have

$$\det(F) = \left(\prod_{s \in S} D_s\right) \cdot \left(\prod_{(s,a) \in S \times A} D_{s,a}\right) \le D^{|S| \cdot |A|},$$

since we have $D_s = 1$ for $s \in S$ and $D_{s,a} \leq D$ for $(s, a) \in S \times A$.

Now, we bound the value of $\det(Id - \mathbf{A})$. First, since $A$ is square matrix, we have $\det(Id - \mathbf{A}) = \det(Id - \mathbf{A}^T)$. Then from (Horn & Johnson, 2012)[Prop. 5.6.P10] we have that

$$\det(Id - \mathbf{A}^T) \leq \prod_{i \in S \cup (S \times A)} ||(Id - \mathbf{A})_{i,.}||_1 .$$

That means we can upper-bound the determinant by the product of sums of absolute values in each row.

But $\mathbf{A}$ is a matrix that describes a stochastic process, which means the sum of absolute values over rows is at most 1. For the identity matrix, we also have that the absolute value of the sum of rows is 1. From the triangle inequality, we have the absolute value of a row of a matrix $Id - \mathbf{A}$ is at most 2.

Since there is $|S| + |S| \cdot |A|$ rows, we have the determinant of the matrix $Id - \mathbf{A}$ is at most $2^{|S| + |S| \cdot |A|}$. That means we have

$$\det(F(Id - \mathbf{A})) \leq (2D)^{|A| \cdot |S|} \cdot 2^{|S|} .$$

So, for any given policy $\pi$, we know that all the reachability values can be expressed as a fraction

$$\frac{p}{q} ,$$

where $p, q \in \mathbb{Z}$ and $q$ is a fixed number for all $s$ (for fixed policy) bounded by $(2D)^{-|A| \cdot |S|} \cdot 2^{-|S|}$.

Observe that immediately implies that the difference between values of two states under one policy is either $0$ or at least $(2D)^{-|A| \cdot |S|} \cdot 2^{-|S|}$.

Now, we examine the reachability value of state $s$ under policy $\pi$ and $\pi'$. We have that $v_s$ can be expressed as $\frac{p}{q}$ and $\frac{p'}{q'}$ for $p, p', q, q' \in \mathbb{Z}$ for policy $\pi$ and $\pi'$ respectively. Moreover, we have $q, q' \leq (2D)^{-|A| \cdot |S|} \cdot 2^{-|S|}$. Then, we can express the difference as

$$\frac{p}{q} - \frac{p'}{q'} = \frac{pq' - p'q}{qq'} .$$

This value is either $0$ or at least $\frac{1}{qq'}$, which is at least $1 / \left( (2D)^{-|A| \cdot |S|} \cdot 2^{-|S|} \right)^2$.

For two policies $\pi, \pi'$, the value of state $s$ is either $0$ or differs by at least

$$(2D)^{-2|A| \cdot |S|} 2^{-2|S|} .$$

$\square$

## G. Practical Modifications

**Choosing $N_k$:** The theoretical value of $N_k$ is astronomically large and cannot be used. Therefore, we use a different value for $N_k$. In the implementation, we have $N_k$ grow quadratically with respect to $k$ and linearly with respect to the number of states in the partial discovered MDP.

This, however, leads to another complication. The $\delta_c$-sure EC algorithm works simply by checking if each potential EC has the right number of samples for the given confidence level. With the implementation's chosen $N_k$, the number of simulations per main-loop iteration grow slower (quadratically) than the number of samples required for $\delta_c$-sure EC detection (super-exponential). Therefore, after a certain iteration of the main loop, the simulation phase of the algorithm will not be able to sample the actions in the EC candidate enough times for the EC to be detected. This will cause BVI to run on a semi-collapsed version of the learned MDP which would not result in the optimal policy. To avoid this optimality loss, we make another change to the algorithm. We do not only check if the required number of samples are taken for EC candidates, but we also run extra samples if they are needed. This way, actual ECs will always be detected.

**Utilizing Heuristic for is_looping Criteria:** The theoretical algorithm's is_looping check runs SCC detection recursively on the staying MDPs found. For large automata and MDPs, this can be very expensive. To quicken this process, we utilize

*Table 1.* Benchmark Performance and Policy Convergence Statistics

| | Description | | | Exploration | | Policy Performance | | | |
|---|---|---|---|---|---|---|---|---|---|
| Benchmark Name | # of States | # of Trans. | % States Seen | % Trans. Seen | Exp. Acc. | Obs. Acc. | Stages Conv. | Samples to Conv. |
| Consensus 2 | 272 | 492 | 100.0 | 98.4 | 1.00 | 1.00 | 2 | 397,467 |
| CSMA 2-2 | 1038 | 1282 | 93.1 | 92.5 | 0.50 | 0.49 | 2 | 569,282,328 |
| Firewire Abst | 623 | 750 | 100.0 | 99.9 | 1.00 | 1.00 | 2 | 3,377,329 |
| IJ 3 | 7 | 21 | 100.0 | 71.4 | 1.00 | 1.00 | 1 | 55 |
| IJ 10 | 1023 | 8960 | 100.0 | 99.8 | 1.00 | 1.00 | 2 | 822,547 |
| Pacman | 498 | 620 | 47.2 | 48.4 | 0.55 | 0.54 | 7 | 9,950,147 |
| Philosophers MDP 3 | 956 | 3696 | 46.0 | 35.4 | 1.00 | 1.00 | 2 | 35,487 |
| Rabin 3 | 27766 | 137802 | 3.9 | 2.0 | 1.00 | 1.00 | 2 | 453,868 |
| Zeroconf | 670 | 997 | 14.2 | 12.9 | 0.00 | 0.00 | 1 | 11,732 |

a heuristic. If the number of states seen in the run does not change within $|S|^2$ samples, it is likely that the simulation is stuck. Therefore, is_looping returns true and the algorithm exits the simulation run. Because we run millions of simulation iterations, it is extremely likely in practice that we will find all relevant transitions.

**Choosing Number of BVI Iterations:** The number of BVI iterations, $n_{BVI}$, should grow with $|S|$ so that all states can propagate their value bounds to the initial state. Through inspection, we found that choosing an $n_{BVI}$ that grows linearly with $|S|$ and $k$ works well. The BVI run converges while also not wasting computing power.

**Convergence Check:** In practice, we want the algorithm to exit rather than run infinitely. Therefore, we have a convergence check. Given the same $\delta_k$, $p_{min}$, and $n_{BVI}$ as the previous iteration, BVI is run on the current iteration. If the error of the initial state's bounds $(U(s_0) - L(s_0))$ is less than that of the previous iteration by more than an input threshold, the algorithm has not converged and continues. Otherwise, it stops. We also added a minimum number of iterations to run the algorithm's main loop since the error of some experiments only improves after a certain number of samples have been seen.

# H. Empirical Results

**MDP PRISM Benchmark Results Table:** The tabular results from the 9 benchmarks the algorithm was tested on can be found in Table 1

**MDP PRISM Benchmark Plots:** Figures 4 to 12 detail the plots for all 9 benchmarks that we tested our algorithm on.

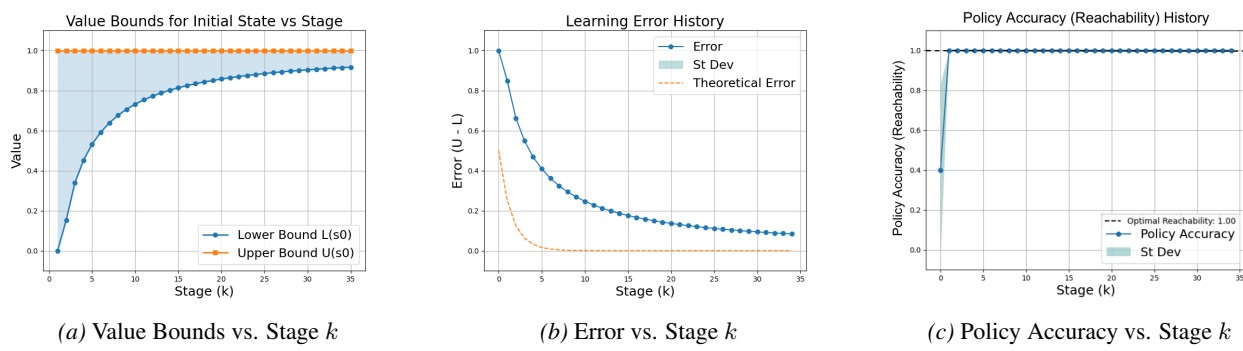

*(a)* Value Bounds vs. Stage $k$    *(b)* Error vs. Stage $k$    *(c)* Policy Accuracy vs. Stage $k$

*Figure 4.* Performance on the Consensus 2 benchmark. The values plotted are the median values over 10 trials. Optimal Policy Reachability Value: 1.00.

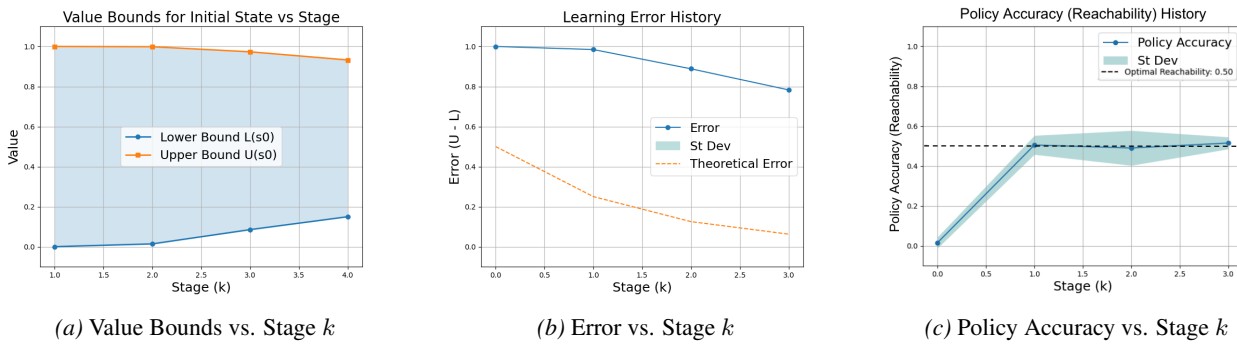

*(a)* Value Bounds vs. Stage $k$      *(b)* Error vs. Stage $k$      *(c)* Policy Accuracy vs. Stage $k$

*Figure 5.* Performance on the CSMA 2-2 benchmark. The values plotted are the median values over 10 trials. Optimal Policy Reachability Value: 0.50.

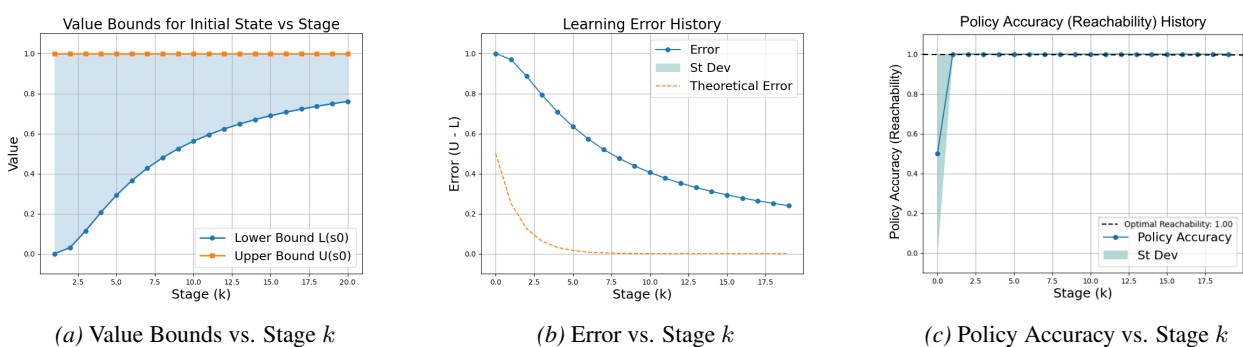

*(a)* Value Bounds vs. Stage $k$      *(b)* Error vs. Stage $k$      *(c)* Policy Accuracy vs. Stage $k$

*Figure 6.* Performance on the Firewire Abst benchmark. The values plotted are the median values over 10 trials. Optimal Policy Reachability Value: 1.00.

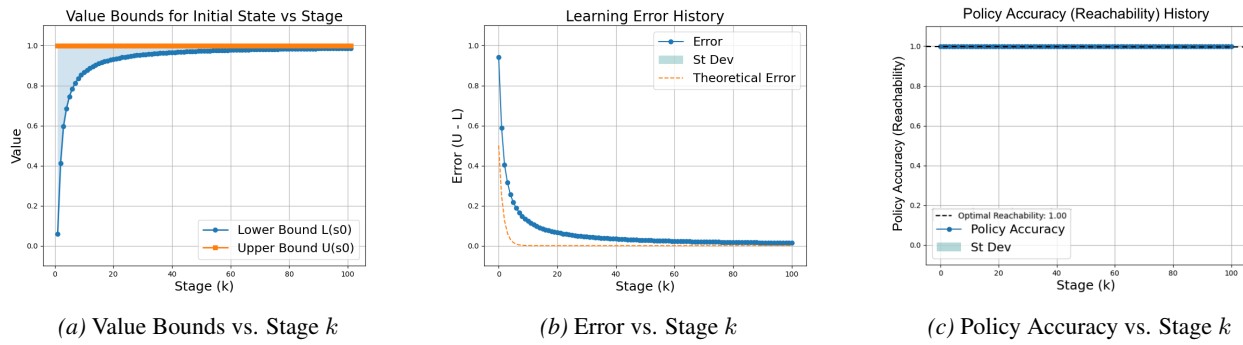

*(a)* Value Bounds vs. Stage $k$      *(b)* Error vs. Stage $k$      *(c)* Policy Accuracy vs. Stage $k$

*Figure 7.* Performance on the IJ 3 benchmark. The values plotted are the median values over 10 trials. Optimal Policy Reachability Value: 1.00.

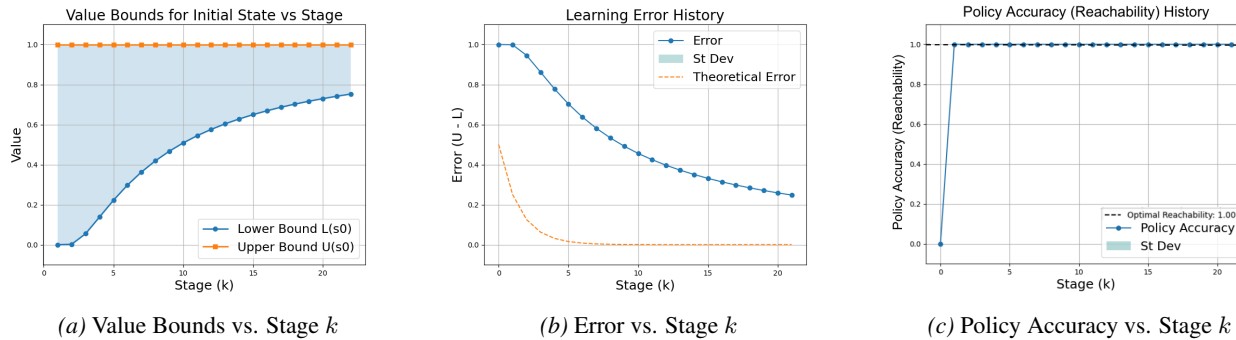

*(a)* Value Bounds vs. Stage $k$      *(b)* Error vs. Stage $k$      *(c)* Policy Accuracy vs. Stage $k$

*Figure 8.* Performance on the IJ 10 benchmark. The values plotted are the median values over 10 trials. Optimal Policy Reachability Value: 1.00.

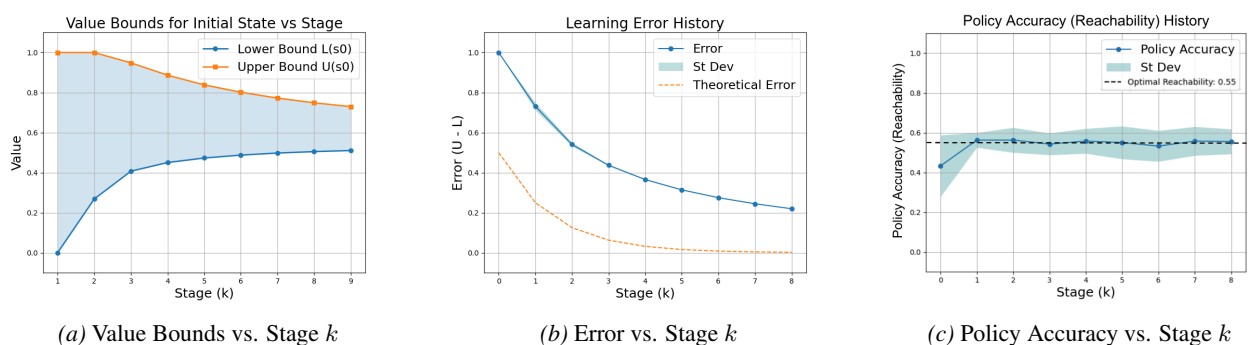

*(a)* Value Bounds vs. Stage $k$      *(b)* Error vs. Stage $k$      *(c)* Policy Accuracy vs. Stage $k$

*Figure 9.* Performance on the Pacman benchmark. The values plotted are the median values over 10 trials. Optimal Policy Reachability Value: 0.55.

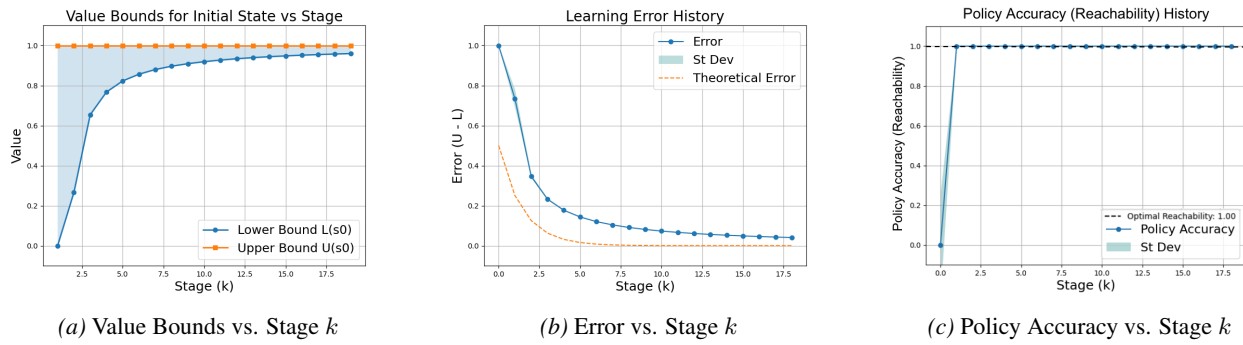

*(a)* Value Bounds vs. Stage $k$      *(b)* Error vs. Stage $k$      *(c)* Policy Accuracy vs. Stage $k$

*Figure 10.* Performance on the Dining Philosophers benchmark. The values plotted are the median values over 10 trials. Optimal Policy Reachability Value: 1.00.

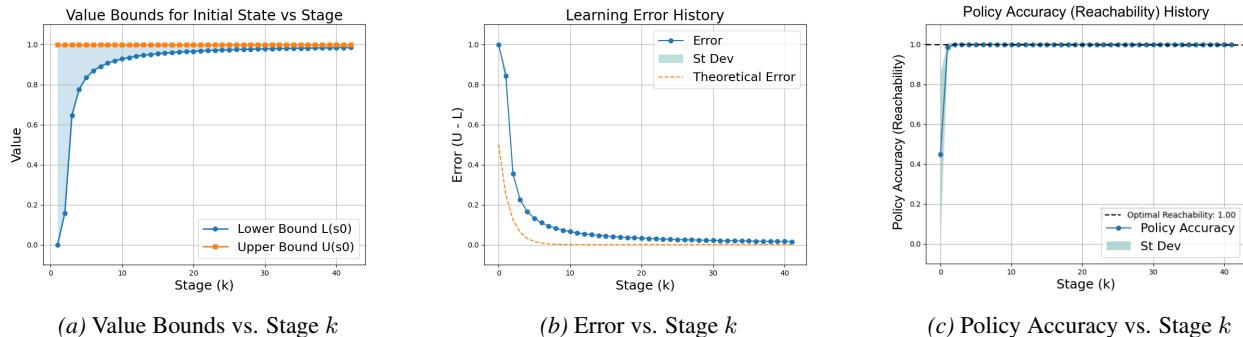

*(a)* Value Bounds vs. Stage $k$      *(b)* Error vs. Stage $k$      *(c)* Policy Accuracy vs. Stage $k$

*Figure 11.* Performance on the Rabin 3 benchmark. The values plotted are the median values over 10 trials. Optimal Policy Reachability Value: 1.00.

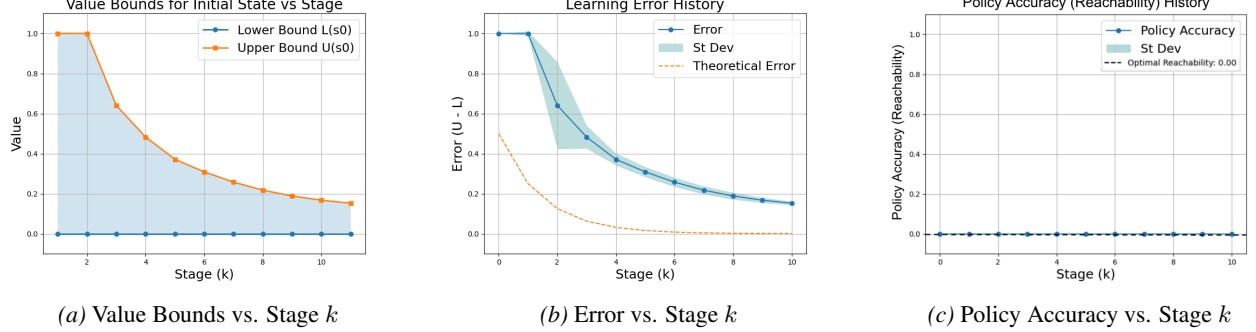

*(a)* Value Bounds vs. Stage $k$      *(b)* Error vs. Stage $k$      *(c)* Policy Accuracy vs. Stage $k$

*Figure 12.* Performance on the Zeroconf benchmark. The values plotted are the median values over 10 trials. Optimal Policy Reachability Value: 0.00.

