# OpenReview forum: "Reinforcement Learning for Reachability: Guaranteeing Asymptotic Optimality"
_ICML.cc/2026/Conference — ICML 2026 regular_

### Official Review · Reviewer_b5Ku · 2026-03-06

**Soundness:** 3
**Presentation:** 3
**Significance:** 2
**Originality:** 3
**Overall Recommendation:** 4
**Confidence:** 3

**Summary:**

This paper studies reinforcement learning for reachability specifications. It builds on PAC (Probably Approximately Correct) learning guarantees and tries to address one important limitation. Standard PAC methods about LTL specification assume that internal MDP parameters are known, such as the minimum transition probability $p_{\min}$, and this value is usually unknown. The authors propose an iterative algorithm that gradually searches for this unknown parameter. At each stage the algorithm makes a guess for a lower bound $p_{\min}$ of the minimum transition probability, using values of the form $p_k = 1/2^k$. At the same time it also reduces the approximation tolerance $\epsilon_k = 1/2^k$ and adjusts the confidence parameter $\delta_k$. The paper proves two main theoretical results. The first result shows that once the guessed bound becomes smaller than the true minimum transition probability, the learned policy is $\epsilon_k$-optimal with probability at least $1 - \delta_k$. The second result shows that after the approximation error becomes smaller than the gap between optimal and suboptimal policies, the algorithm will only produce optimal policies with probability one. The algorithm itself is based on bounded value iteration (BVI). It operates on a collapsed MDP and uses end component detection to deal with states where the agent may become trapped.

**Compliance With Llm Reviewing Policy:**

Affirmed.

**Final Justification:**

I give this paper a Weak Accept. The paper studies an interesting problem and has clear originality and potential significance. My main concerns were about, especially the role of some unknown quantities in the theory, the effect of EC detection errors, and the practical stopping criterion.

After reading the authors’ rebuttal, I am convinced that these concerns were addressed well. In particular, the rebuttal clearly explained that the questioned quantities are only used for theoretical analysis and are not required as inputs to the algorithm. The authors also clarified why EC detection errors do not break the overall guarantee across stages. These responses resolved my previous confusion.

I am staying a weak accept though because I am unsure about how significant the result is.

**Key Questions For Authors:**

1. The bound for $s_k$ on line 992 implies exponential growth in problem size. What is the practical implication for even moderately sized MDPs? Can you provide empirical sample complexity numbers?

2. In Figure 1-type scenarios where escape probability from an EC is $\epsilon << p_k$, how does the algorithm avoid incorrectly classifying transient regions as end components? What happens when $\epsilon_{diff}$ is comparable to rare escape probabilities?

4. I think there is a gap between theoretical and practical EC detection. For example, the theoretical EC detection is correct only when two conditions hold: (1) $k \ge K_{\text{PAC}}$ so that $p_k \le p_{\min}$, and (2) the required number of samples  $\frac{\ln(\delta_C)}{\ln(1-p_k)}$ is collected. However, both conditions appear difficult to satisfy in practice. Because the threshold $K_{\text{PAC}}$ depends on $p_{\min}$, which is unknown, so practitioners cannot know when the condition $p_k \le p_{\min}$ actually holds. Also, during stages where either condition is not satisfied, false EC detection may produce incorrect policies. Since simulations in stage $k+1$ are guided by the policy learned in stage $k$, early mistakes may propagate and cause the algorithm to miss important transitions. Does this remove the theoretical guarantees? What empirical evidence supports that the heuristic implementation still converges to optimal policies, especially for benchmarks such as Pacman (policy value 0.55) and Zeroconf (policy value 0.00) where convergence appears to fail?

3. Since $K_{PAC}$ and $K_{opt}$ depend on unknown quantities, what practical stopping criterion does the algorithm use? How would a user know when optimality is achieved?

5. The proof requires $\epsilon_k < \epsilon_{diff}$. However, $\epsilon_{diff}$ could be arbitrarily small. Does this affect the practical applicability of the result?

6. Line 23 has a citation link error in the second column.

I could've been generous for my evaluation, if the limitations i described are transparently described as limitations (meanwhile, I am aware of my evaluation could've been wrong), also explaining some insights why some benchmark results failed. Currently, I feel like the paper is hiding all the weaknesses. But I am willing to raise my score if my questions are properly addressed.

**Limitations:**

partially yes, they only mention some of limitations like model-based only hiding another unknown quantity of $K_{PAC}$ and $K_{opt}$ which needs for stopping criterion. I feel like this result just converts one unknown value into another unknown values. Also, lack of explanation why some of benchmark results look so bad for example, Pacman and Zeroconf environments.

**Strengths And Weaknesses:**

**Strengths**
1. **Addresses a fundamental gap**: The paper identifies that existing asymptotic convergence guarantees provide limited insight into convergence dynamics. The PAC-style analysis offers more explicit bounds. To address the gap, the idea of systematically lowering $p_k$ to eventually satisfy unknown PAC conditions is detectable (as $p_k$ goes to 0, it surely happens that some step, it's smaller than $p_{\min}$).

**Weaknesses**
1. **Hidden assumption on $p_k$ convergence**: The paper sets $p_k = 1/2^k$ so that it will eventually satisfy $p_k ≤ p_{min}$ since $1/2^k$ goes to 0 as $k$ goes to $\infty$. While mathematically sound, this hides a critical practical issue of the sample complexity $s_k$ grows as $(1/p_k)^{|S_C|·r}$, which becomes exponentially large as $p_k$ decreases. The paper does not adequately discuss this exponential explosion.

2. **Non-constructive thresholds**: Both $K_{PAC}$ and $K_{opt}$ are existence results, they provably exist but cannot be computed without knowing $p_{\min}$ and $ε_{diff}$. The paper does not explain how a practitioner would know when the algorithm has reached these thresholds.

3. **Looping detection flaw**: The $\delta_C$-SURE-EC test (Algorithm 4) determines if a state set is an end component by sampling. However, if the true escape probability from an flagged as EC (but it actually not because it might have very small chance of escaping) is extremely small (e.g., $\epsilon = 10^{-8}$), the algorithm may incorrectly conclude the agent is "stuck" in a loop. The paper provides no analysis of false positive EC detection when escape probabilities are rare but non-zero.

---

> ### Author Rebuttal · Authors · 2026-03-28
>
> Dear Reviewer b5Ku, thank you for your review.
>
> In summary, our response addresses your key concerns. We affirm the theoretical guarantees hold despite the $\delta_{EC}$ error in EC detection (please see responses to Q5, W3, and Q2). We are not replacing the unknown $p_{min}$ with other unknowns such as $K_{PAC}$ or $K_{OPT}$; these parameters are only analytical (required for theoretical analysis). They are never input into the algorithm (please see response to Q4 and W2). The algorithm performs well on all benchmarks (please see response to Q3). Finally, practitioners would treat this algorithm's termination like every other algorithm with asymptotic guarantees (e.g. Q-learning). Please see detailed response to all weaknesses and questions below:
>
> [In response to Q1 + W1] The bound on $s_k$ is exponential, but this is a lower bound. All algorithms solving reachability require an exponential number of samples to obtain theoretical guarantees. Hence, this is not a limitation of our approach per se. On the contrary, we would argue that our algorithm's choice of exponentially decaying the parameters $p_k$, $\varepsilon_k$, and $\delta_k$ actually speeds up the process. Note that other than the exponential decay rate of $\delta_k$, the other decay rates could have been even slower for the theoretical guarantee to hold. Our implementation uses significantly fewer samples than $s_k$ (and $N_k$): we use $10 \cdot k^2 \cdot (|S|+1)$ samples for stage k. Sample complexity is mentioned in Appendix G. For specific values, please see the “Samples to Conv.” column in Table 1 in Appendix H. Also, the algorithm on average converges in less than 3 stages (2.3 stages as mentioned in Section 5).
>
> [In response to Q4 + W2] Parameters $K_{PAC}$ and $K_{OPT}$ are not parameters for the functioning of the algorithm. They do not appear in Algorithm 1. They are simply parameters needed to theoretically analyze algorithm correctness. Thus, we're not replacing one algorithmic unknown with another. In practice, like every other asymptotically converging algorithm, such as Q-learning, practitioners terminate when the policy’s performance converges (i.e. there is minimal improvement between the previous and current stage). For us, performance is computed as the probability with which the policy reaches the target set, i.e. policy accuracy: graph C of Figures 4-12.
>
> [In response to Q5] $\varepsilon_k < \varepsilon_{diff}$ is needed for the theoretical guarantee. It implies the number of stages required to ensure the guarantee holds. In practice, the number of stages required before policy convergence was less than 3 (as mentioned in Section 5), indicating a large gap between the theoretical analysis and practical performance. This warrants a tighter theoretical analysis in the future.
>
> [In response to W3 and Q2] Our EC detection algorithm is described in Algorithm 4 of the appendix. The guarantee is probabilistic; it may classify any EC incorrectly (the error is subsumed within $\delta_{EC}$). This is part of the theoretical guarantee. For further details, please refer to Appendix A1, C, and E. As to why incorrect collapses in stage k don't propagate errors into stage k+1, please see our response to [Q3].
>
> [In response to Q3] We answer each part of this question separately.
>
> Regarding $K_{PAC}$ and $p_{min}$, please refer to our response to [Q4 + W2].
>
> Regarding false EC detection (continuing from response to [W3 and Q2]), we explain why EC misclassifications in stage k do not affect overall theoretical guarantees. The algorithm only utilizes upper bounds from the previous stage's BVI run as a guiding heuristic for simulation. Every stage, bounds are recomputed from scratch; we'll emphasize this in the final version. For collapsed ECs, the upper bounds of the state-action (SA) pairs that made up the EC are equal to the upper bound of the EC’s super-state during BVI (i.e. upper bound of EC’s best exit action). Note, these upper bounds may be inflated but will still be valid, meaning incorrectly collapsing an EC doesn't propagate error into the next stage. Moreover, as sampling goes on, the bounds on estimated transition probabilities tighten. Eventually, BVI upper value bounds of states will converge to the true values. The algorithm samples the SA pairs with the highest upper bounds. Because upper bounds are correct and only decrease, the SA pairs with the true highest values will eventually be found. Thus, errors from previous stages (including $\delta_{EC}$ error of incorrectly collapsing ECs) don't affect the algorithm’s convergence guarantees.
>
> Regarding performance on PACMAN and Zeroconf, please note the true reachability probability (policy accuracy) of the optimal policies in these benchmarks is 0.55 and 0, respectively. Hence, our implementation's performance on these benchmarks is not indicative of failure. We converged to 0.54 and 0 within 7 and 1 stages on these benchmarks, respectively. Please see Table 1 for details.

---

> > ### Author Rebuttal · Reviewer_b5Ku · 2026-04-02
> >
> > Thank you for the detailed rebuttal. I am now fully convinced by the authors’ clarifications. They resolved my previous confusion very well, especially regarding the role of the analytical quantities in the theory versus the actual algorithm, the handling of EC detection errors, and why such errors do not invalidate the convergence guarantee across stages.
> >
> > Overall, the rebuttal addressed my concerns clearly and satisfactorily. Accordingly, I am raising my score.

---

> > > ### Author Response · Authors · 2026-04-02
> > >
> > > Thank you for reviewing our work!

---

### Official Review · Reviewer_HXtW · 2026-03-12

**Soundness:** 4
**Presentation:** 3
**Significance:** 4
**Originality:** 4
**Overall Recommendation:** 5
**Confidence:** 2

**Summary:**

The paper under review studies reachability of finite Markov decision processes. Particularly, given a set of target states $T$ in a MDP, it deals with the computation of the policy whose probability of visiting states from $T$ is highest. They construct an Algorithm that produces at $k$-stage an policy whose value-function is $\epsilon_k$ apart from the value function of the optimal policy. They essentially show that there exists $\epsilon_{\mathrm{diff}}>0$ such that the $\epsilon_{\mathrm{diff}}$-ball around any policy only contains the policy itself, and hence the algorithm converges to the exact optimal policy as at some point $\epsilon_k<\epsilon_{\mathrm{diff}}$. The constant $\epsilon_{\mathrm{diff}}$ depends on parameters of the MDP and in Section 4, the authors show that it can be bounded in the case that the transition probabilites are rational numbers.

**Compliance With Llm Reviewing Policy:**

Affirmed.

**Final Justification:**

Overall, find the results of the paper very convincing, mathematically sound, and sufficiently novel. I initially gave this paper an **Accept** and raised only minor remarks and questions in my review that all have been answered by the authors. Thus, I am maintaining my "Accept".

**Key Questions For Authors:**

see above.

**Limitations:**

yes

**Strengths And Weaknesses:**

I find the results of the paper of high importance and very valuable for the RL community. The mathematical approach to proof the convergence bound seem correct. The paper is very well written and mathematically precise.
Although having a focus on theory, the paper also comes with empirical validation showing the practicability of the results. I only have some minor comments, questions, and suggestions:

- An essential observation is that $\epsilon_{\mathrm{diff}}>0$ holds true. In the proof of Theorem 3.2, its said this is true because "there are only finitely many policy". There, it should be added that this is true for memoryless and deterministic policies, as $\Pi(S, A)$ is not finite in general. As the definition of this constant is essential, it recomment to not hide it in the proof but to put in a proper definition environment.
- I wonder how much theoreetical and practical insight the restriction to MDPs having rational transition probabilities really gives, because it simply replaces one unkown constant, namely $\epsilon_{\mathrm{diff}}$, with another unkown $D$. Ultimatively, I  think  $\epsilon_{\mathrm{diff}}$ only depends on matrix properties of $A$ and the target set $T$, right? For instance, is it possible to relate $\epsilon_{\mathrm{diff}}$, like the conductance of $T$ in the Markov chain induced by $A$?
- In the defintion of $J_L^M$, isn't the input to $D_\pi^\mathcal{M}$ just $\mathrm{Runs}(S, A)\cap \mathcal{L}$, or simply $\mathcal{L}$ because its a subest of all runs?
- The preliminaries could be improved for the RL community of ICML by mentioning very early, like where MDPs are defined, that there is a certain target set $T$ that needs to be reached and that reward signals are trivial here. An equivalent definition is probably the one from Section 4 with reward vector $r$ and $\gamma=1$? There, should $w_s$ for $s\in T$ be set to $1$ or what is the value here?
- How exactly does `X ← SIMULATE(M, T )` (Line 9 in Algorithm 1) work? Why does this not affect convergence of the transition probability estimation?

---

> ### Author Rebuttal · Authors · 2026-03-27
>
> Dear Reviewer HXtW, thank you for your review.
>
> Thanks for the comment. We will explicitly state that the policies are memoryless and deterministic in Theorem 3.2.
>
> To clarify, rationality is not needed for Section 3. There, only the argument that there are finitely many (memoryless deterministic) policies suffices. In Section 4, with the natural assumption on rationality of transitions, we quantify the gap between two policies in terms of parameters of the MDP. The conductance and mixing times point in the right direction; they can be used to bound the difference between two states in one Markov Chain. In MDPs, the problem is a little trickier since we need to bound the difference between two different Markov Chains.
>
>
> In the definition of $J_M^L(\pi)$, the notation is to represent all trajectories produced by the policy $\pi$ in the MDP M which satisfy the specification L. The notation of [ $Runs(S,A)\cap L$ ] or L can include trajectories that are not produced by $\pi$ in M.
>
> We will incorporate your suggestions into the preliminaries in the final version.
>
> The SIMULATE algorithm is described in Algorithm 2 in the appendix. It starts the agent at $s_0$, then at every state an action is chosen randomly from the best actions for that state (i.e., the actions with the highest upper bounds on expected reachability for that state), lastly the current state is updated based on the transition probability taken by the chosen action. The simulation terminates if a goal state is reached or if the algorithm detects that it is stuck in an EC. This method ensures that all relevant actions and transitions have a non-zero chance of being sampled.
> Because of our choice of $N_k$, as mentioned in the proof of Theorem 3.1, we can guarantee with high probability that all relevant actions are chosen at least $s_k$ times. Hence, the convergence guarantees hold using this simulation function.

---

> > ### Author Rebuttal · Reviewer_HXtW · 2026-04-01
> >
> > Thank your for the clarification!

---

> > > ### Author Response · Authors · 2026-04-02
> > >
> > > Thank you for reviewing our work!

---

### Official Review · Reviewer_h2xN · 2026-03-12

**Soundness:** 4
**Presentation:** 3
**Significance:** 3
**Originality:** 4
**Overall Recommendation:** 4
**Confidence:** 3

**Summary:**

The paper proposes a novel algorithm to address LTL (linear temporal logic) RL problems with PAC guarantees. It provides the theoretical analysis (in particular PAC proofs in Sec 3) as well as empirical studies to validate the theoretical results.

**Compliance With Llm Reviewing Policy:**

Affirmed.

**Final Justification:**

The clarifications in the rebuttal were helpful. As in the original review, I think this is a solid and interesting theory paper on RL for LTL.

**Key Questions For Authors:**

I mentioned above: I would be interested to learn more about the relation to the (classical/old) theoretical analysis of polynomial time near-optimal RL (Rmax or E^3 or KWIK).

**Limitations:**

Only the model-based approach is mentioned as limitation. As this is a theory focused paper, this is fine: The theorems are true, and that's it.

**Strengths And Weaknesses:**

* First, from the title and abstract it is not clear that RL maximizing expected success under LTL specifications (Eq. page 2) is addressed. The title and abstract only mentions 'RL for reachability' -- I was not aware and would be confused if this is meant to mean "RL with temporal logic".

* Not being an expert in LTL settings, my questions revolved mostly around relation to guaranteed RL in standard reward settings: Rmax achieves approximate optimality in polynomial time; I believe KWIK RL similarly. In my understanding your results "only" provide asymptotic near-optimality. Is there a fundamental reason why the LTL case is harder (and harder to theoretically analyze) than standard RL? Are there relations between the analysis techniques in Rmax (and say, E^3 (paper title "near-optimal RL in polynomial time")). These seem to consider bounds on the estimated transitions similar to this paper. Are there relations (or have such relations previously been discussed in literature you already mention?) Generally, it would help to 'cross-relate' the existing theoretical work on standard reward RL with that on LTL RL.

* Generally, the idea of iteratively reducing an assumed p_{min} seems intuitive and original, and the resulting algorithm simple and insightful.

---

> ### Author Rebuttal · Authors · 2026-03-27
>
> Dear Reviewer h2xN, thank you for your review.
>
> Regarding our mention of RL for reachability vs. RL for linear temporal logic (LTL), our algorithm specifically solves the reachability problem. Reachability tasks are a strict subset of LTL. However, in MDPs, LTL objectives can be converted to reachability objectives for optimal policies, as explained in the Remark 2.1 and the related work. Hence, our research is a crucial step to solving RL for LTL.
>
> Regarding the use of standard discounted-sum RL, please refer to the General Comment GC1 under Reviewer ZQjn. In short, there exists no translation between reachability to discounted-sum RL that preserves optimal or near-optimal behaviors. Thus, we cannot directly compare or relate our LTL RL method with reward-based RL methods like Rmax, E^3, or KWIK.

---

> > ### Author Rebuttal · Reviewer_h2xN · 2026-04-02
> >
> > Thanks for the clarifications.

---

> > > ### Author Response · Authors · 2026-04-02
> > >
> > > Thank you for reviewing our work!

---

### Official Review · Reviewer_ZQjn · 2026-03-13

**Soundness:** 4
**Presentation:** 3
**Significance:** 3
**Originality:** 3
**Overall Recommendation:** 5
**Confidence:** 3

**Summary:**

The paper considers reinforcement learning for reachability objectives in unknown MDPs. It proposes a staged algorithm that combines progressively refined guesses of unknown MDP parameters, partial-model estimation, and bounded value iteration, and proves that with probability one there is a finite stage beyond which the algorithm outputs only optimal policies.

**Compliance With Llm Reviewing Policy:**

Affirmed.

**Final Justification:**

After reading the authors' rebuttal and the discussion with the other reviewers, I see that my assessment is aligned with theirs, and I choose to keep my score.

**Key Questions For Authors:**

Since the target set of states $T$ is assumed to be known, could the problem be solved and the same type of guarantees be possibly obtained by applying more standard RL methods after assigning reward of 1 to the states in $T$ and 0 anywhere else, as is commonly done in goal-conditioned RL?

**Limitations:**

The limitations of this theoretical work are clearly discussed in the paper. I also do not see potential negative societal impact specific to this work.

**Strengths And Weaknesses:**

The paper proposes a new algorithm for solving the reachability problem with a PAC guarantee, and provides its theoretical analysis. The algorithm itself is fairly straightforward and builds on previously developed ideas, such as bounded value iteration and collapsing, but the overall combination is meaningful and well motivated for the problem considered.

In my view, the paper is clear and well written. The presentation is easy to follow, the technical development appears sound, and I found the paper pleasant to read. Another positive aspect is that the theoretical claims are supported by an experimental evaluation, which helps connect the analysis to the practical behavior of the method.

One small issue is the notation. I would suggest that the authors do not use $T$ for the set of target states, nor $t$ for the next state, since this may be confused with the time step. Using $G$ for the target set and $s'$ or $g$ for states might be clearer, as is common in goal-conditioned RL.

Overall, my impression is positive. While the algorithmic ingredients are not entirely new in isolation, the paper puts them together in a coherent way to address an interesting problem, and the resulting contribution appears technically solid and convincing.

---

> ### Author Rebuttal · Authors · 2026-03-27
>
> Dear Reviewer ZQjn, thank you for your review.
>
> We will incorporate all editorial suggestions.
>
> Regarding the use of trivial rewards, please refer to the General Comment GC1 below:
>
> GC1. Comparison to standard discounted-sum based RL/goal-conditioned RL/E^3 etc/RMAX:
>
> The intractability framework of [Alur et. al. 2022] establishes that there is NO optimal- or near-optimality-preserving translation from reachability objectives into discounted sum-rewards in RL, ruling out any reduction between the two problem classes. Reachability optimizes the probability of reaching a target state: In terms of rewards, a trajectory receives reward 1 if it visits the target and 0 otherwise. Even with these trivial rewards, discounted-sum based RL (including the likes of standard RL/E^3 algorithm, goal-conditioned RL) will assign a reward of $1/\gamma^k$ to a trajectory that visits the target state at the k-th position for the first time, and 0 otherwise where $0<\gamma<1$ is the discount-factor. We will add this discussion to the related work section in the final version.
>
> (Le et al. 2024) attempt to use goal-conditioned RL with discounted-sum rewards by reducing reachability to an infinite sequence of discounted-sum RL problems. However, as noted in the paper, their reduction admits only limited theoretical analysis (the work of Le et al. requires external parameters).

---

> > ### Author Rebuttal · Reviewer_ZQjn · 2026-04-03
> >
> > The suggestions from my review have been incorporated into the paper, and my questions have been fully answered.

---

> > > ### Author Response · Authors · 2026-04-03
> > >
> > > Thank you for reviewing our work!

---

### Decision · Program_Chairs · 2026-04-30

**Decision:**

Accept (regular)

**Comment:**

Reviewers generally appreciated the work and mentioned that the work is valuable to the RL community, particularly researchers working at the intersection of LTL and RL. The theoretical results seems sound and the simulation results are a plus. I suggest the authors to include some of the discussion pointers raised by the reviewers.